# A novel bacterial effector protein mediates ER-LD membrane contacts to regulate host lipid droplets

Rajendra Kumar Angara (ID), Arif Sadi (ID) & Stacey D Gilk (ID) ✉

## Abstract

Effective intracellular communication between cellular organelles occurs at dedicated membrane contact sites (MCSs). Tether proteins are responsible for the establishment of MCSs, enabling direct communication between organelles to ensure organelle function and host cell homeostasis. While recent research has identified tether proteins in several bacterial pathogens, their functions have predominantly been associated with mediating inter-organelle communication between the bacteria containing vacuole (BCV) and the host endoplasmic reticulum (ER). Here, we identify a novel bacterial effector protein, *Cb*EPF1, which acts as a molecular tether beyond the confines of the BCV and facilitates interactions between host cell organelles. *Coxiella burnetii*, an obligate intracellular bacterial pathogen, encodes the FFAT motif-containing protein *Cb*EPF1 which localizes to host lipid droplets (LDs). *Cb*EPF1 establishes inter-organelle contact sites between host LDs and the ER through its interactions with VAP family proteins. Intriguingly, *Cb*EPF1 modulates growth of host LDs in a FFAT motif-dependent manner. These findings highlight the potential for bacterial effector proteins to impact host cellular homeostasis by manipulating inter-organelle communication beyond conventional BCVs.

**Keywords** Inter-organelle Contacts; FFAT Motif; Lipid Droplets; Molecular Tethers; *Coxiella burnetii*
**Subject Categories** Membranes & Trafficking; Microbiology, Virology & Host Pathogen Interaction; Organelles

## Introduction

Inter-organelle communication, the dynamic exchange of biomolecules, ions, and lipids between cellular organelles, is integral to maintaining cellular homeostasis. In the past decade, extensive research has unveiled inter-organelle membrane contact sites where key organelles, including the plasma membrane, endoplasmic reticulum (ER), mitochondria, Golgi apparatus, peroxisomes, and lipid droplets (LDs), engage in direct communication (Rizzuto et al, 1998; Hanada et al, 2003; Szymanski et al, 2007; Giordano et al, 2013; Peretti et al, 2008; Friedman et al, 2013; Costello et al, 2017; Hua et al, 2017; Wu et al, 2018; Alpy et al, 2013; Scorrano et al, 2019). Inter-organelle contact sites are orchestrated by unique proteins with distinct motifs and domains that act as molecular tethers between organelles and facilitate biomolecule exchange (Eisenberg-Bord et al, 2016; Gatta and Levine, 2017; Silva et al, 2020). These sites serve as hubs for lipid transfer, calcium signaling, and membrane trafficking, and profoundly influence cellular processes such as lipid metabolism, signal transduction, and organelle biogenesis (Zajac et al, 2013; Prinz, 2014; Hirabayashi et al, 2017; Wu et al, 2018; Bohnert, 2020; Prinz et al, 2020; Silva et al, 2020).

Pathogens employ intricate strategies to co-opt host cell machinery and manipulate cellular metabolism to ensure their survival and replication. Recent studies have revealed the pathogen's ability to modify host inter-organelle communication (Ettayebi and Hardy, 2003; Hamamoto et al, 2005; Derré et al, 2011; Agaisse and Derré, 2015; Justis et al, 2017; Kors et al, 2022; Jiang et al, 2022; Yue et al, 2023). Upon host cell entry, certain obligate intracellular bacteria adopt a unique membrane-bound vacuole, referred to as the Bacteria-Containing Vacuole (BCV), to evade the host immune response and support bacterial replication (Santos and Enninga, 2016; Petit and Lebreton, 2022). While the BCV shields bacteria, it restricts access to host cellular machinery and nutrients. Consequently, various types of secretion systems have evolved in different bacteria to facilitate delivery of bacterial effector proteins into host cells (Mitchell et al, 2016). Bacterial effectors localize to various host cell organelles, using molecular mimicry to manipulate host metabolism and promote bacterial survival. Recent studies have unveiled that, in addition to adopting eukaryotic catalytic and receptor domains as a form of molecular mimicry, bacterial effector proteins can also contain features of eukaryotic tether proteins and establish inter-organelle contact sites between the host ER and BCV membranes (Murray et al, 2017; Vormittag et al, 2023). Remarkably, although several bacterial effector proteins localize to host cell organelles outside the context of the BCV, none have been identified to mediate inter-organelle contact sites.

The ER serves as a central hub at the heart of inter-organelle communication (Phillips and Voeltz, 2016; Wu et al, 2018; Almeida and Amaral, 2020). The majority of ER-driven inter-organelle communication is facilitated by FFAT (two phenylalanines in an acidic tract) motifs found in a diverse family of proteins (Murphy and Levine, 2016). The FFAT motif-containing proteins act as molecular tethers between ER-resident VAPs (VAMP-associated proteins) and various organelles (Neefjes and Cabukusta, 2021). Besides their role as tethers, FFAT motif-containing proteins can also function as lipid or ion transporters. While considered a eukaryotic motif, FFAT motifs are also found in secreted bacterial effector proteins which mediate inter-organelle contact sites between the ER and BCVs (Murray et al, 2017). Given that a

Department of Pathology, Microbiology, and Immunology, University of Nebraska Medical Center, Omaha, NE, USA. ✉E-mail: sgilk@unmc.edu

substantial portion of intracellular bacteria effector proteins remain uncharacterized, it is plausible that many unidentified FFAT motif-containing effector proteins localize to host cell organelles and facilitate host inter-organelle communication.

Building upon existing knowledge of inter-organelle communication, our study investigates a unique *Coxiella burnetii* effector protein that participates in inter-organelle communication beyond BCVs. The *C. burnetii* effector protein containing FFAT motifs, termed *Cb*EPF1, establishes inter-organelle contact sites between the host ER and LDs. *Cb*EPF1 features two FFAT motifs, each demonstrating preferential interaction with VAPs in the ER. *Cb*EPF1 mediates ER-LD inter-organelle contact sites to promote development of larger LDs in an FFAT motif-dependent manner, offering insights into how *C. burnetii* reprograms host lipid metabolism.

## Results

### *Cb*EPF1 localizes to host ER and LDs

To identify effector proteins that may establish inter-organelle contact sites beyond BCVs, we conducted a bioinformatic screening of 127 predicted *C. burnetii* Type 4B Secretion System (T4BSS) effector proteins (Chen et al, 2010; Bi et al, 2013) for FFAT motifs using FIMO (Find Individual Motif Occurrences) (Grant et al, 2011). From this screen, CBU1370 contained the highest probability with two FFAT motif sequences, both bearing strong similarity to the consensus sequences and preceded by acidic amino acids (Fig. 1A) (Neefjes and Cabukusta, 2021). Previous studies confirmed CBU1370 as T4BSS effector protein (Lifshitz et al, 2014). Beyond the predicted FFAT motifs, the CBU1370 protein sequence contains a putative amphipathic helix at the C-terminal end (from 262–279 amino acids). Accordingly, we designated CBU1370 as *Coxiella burnetii* Effector Protein with FFAT motifs 1 (*Cb*EPF1).

To understand the functional significance of *Cb*EPF1 in host cells, we ectopically expressed *Cb*EPF1-GFP in mCherry-*C. burnetii* infected epithelial cells. *Cb*EPF1-GFP predominantly localizes to the host ER with enrichment at distinct loci within the host cytosol (Fig. 1B top panel). Using co-localization with several organelle and vesicular markers, we determined that *Cb*EPF1-GFP also colocalizes with host LDs (Fig. 1B bottom panel). As LD biogenesis occurs at the ER (Kassan et al, 2013; Choudhary et al, 2015), we next used live cell imaging to examine the spatial and temporal localization of *Cb*EPF1 during LD biogenesis. We found that *Cb*EPF1-GFP initially localizes to ER sites of LD biogenesis and gradually enriches on the LD surface as LDs grow (Fig. 1C,D). Overall, these results demonstrate that *Cb*EPF1 dynamically distributes between the ER, LD biogenesis sites and the LD surface.

### *Cb*EPF1 relocates from ER to the LD surface and induces inter-organelle contact sites between the host ER and LDs

To gain deeper insight into *Cb*EPF1 localization dynamics at the ER and LDs, we induced LD synthesis in HeLa cells by supplementing the media with oleic acid (OA). As LD number increased over time, we observed a corresponding decrease in *Cb*EPF1-GFP associated with the ER (Fig. 2A). In tandem, there was noticeable enrichment of *Cb*EPF1-GFP on host LD surfaces. This suggests a dynamic

translocation of *Cb*EPF1-GFP from the ER to the LD surface, a process that coincides with the increase in LD size (Fig. 2A). Several eukaryotic proteins implicated in LD biogenesis or growth exhibit similar translocation pattern from the ER to LD surface as the LD grow to larger size (Wilfling et al, 2013; Xu et al, 2012; Chung et al, 2019; Li et al, 2019). While larger LDs often detach from the ER and are released into the cytosol, they can also maintain contact with the ER by other mechanisms (Hugenroth and Bohnert, 2020). In our studies, cells expressing *Cb*EPF1-GFP presented a distinctive pattern, with the ER enveloping the *Cb*EPF1-GFP-localized LDs. The ER exhibited notably increased and extended association around *Cb*EPF1-GFP-localized LDs (Fig. 2B,Cii). In contrast, in control GFP-expressing cells the ER forms a reticulate network with only limited association with LDs (Fig. 2B,Ci). Remarkably, the ER wrapping around *Cb*EPF1-localized LDs was not always continuous but was distinctly confined to regions where *Cb*EPF1 localized on the LDs (Figs. 2B,Cii and EV1). This indicates that *Cb*EPF1 initiates and mediates contact sites between the ER and *Cb*EPF1-localized LDs, emphasizing its significance in orchestrating inter-organelle interactions.

### *Cb*EPF1 contains two FFAT motifs and interacts with ER VAP proteins

To understand the factors responsible for *Cb*EPF1-mediated ER-LD contact sites, we focused on the *Cb*EPF1 FFAT motifs. VAPs in the ER membrane form heterodimeric complexes and establish inter-organelle contact sites between the ER and various organelles (Fig. 3A) (Murphy and Levine, 2016). VAPs, specifically VAPA, VAPB, and MOSPD2, form inter-organelle contact sites through the VAP MSP domain which binds the FFAT motif on partner proteins found on other organelles (Cabukusta et al, 2020). Consequently, proteins harboring FFAT motifs typically act as molecular tethers at inter-organelle contact sites. To evaluate whether *Cb*EPF1 binding to VAPs is responsible for *Cb*EPF1-mediated ER-LD contacts, we used an in vitro bacterial adenylate cyclase-based two-hybrid assay (BACTH) to test for and interaction between *Cb*EPF1 and VAPB. In the BACTH assay, *Cb*EPF1 interacted with wild-type VAPB but not with a mutant VAPB containing a mutated MSP domain (VAPB-MSPmt), the VAP domain critical for interaction with FFAT motif-containing proteins (Fig. 3C). To determine whether the *Cb*EPF1 FFAT motifs are required for binding to VAPB, we generated mutations in the two predicted *Cb*EPF1 FFAT motifs. While amino acid substitutions at most positions within the FFAT motif do not affect function, the phenylalanine or tyrosine residue at position 2 is indispensable for effective VAP binding (Murphy and Levine, 2016). Therefore, to test the functional importance of the putative *Cb*EPF1 FFAT motifs, we introduced alanine substitutions in position 2 of the first FFAT motif (Y231A, referred as F1mt), the second FFAT motif (F244A, referred as F2 mt), or both FFAT motifs (Y231A-F244A, referred to as YF/AA or F3mt) (Fig. 3B). In the BACTH assay, wild-type VAPB (VAPB-Wt) interacted with *Cb*EPF1-F1mt and *Cb*EPF1-F2mt but not with *Cb*EPF1-F3mt (Fig. 3C). These results indicate that *Cb*EPF1 interacts with VAPB through the FFAT and MSP domains. Further, while both *Cb*EPF1 FFAT motifs can bind VAPB, at least one functional FFAT motif is required and sufficient for *Cb*EPF1-VAPB binding. While no additional *C. burnetii* effector or eukaryotic factors are required for

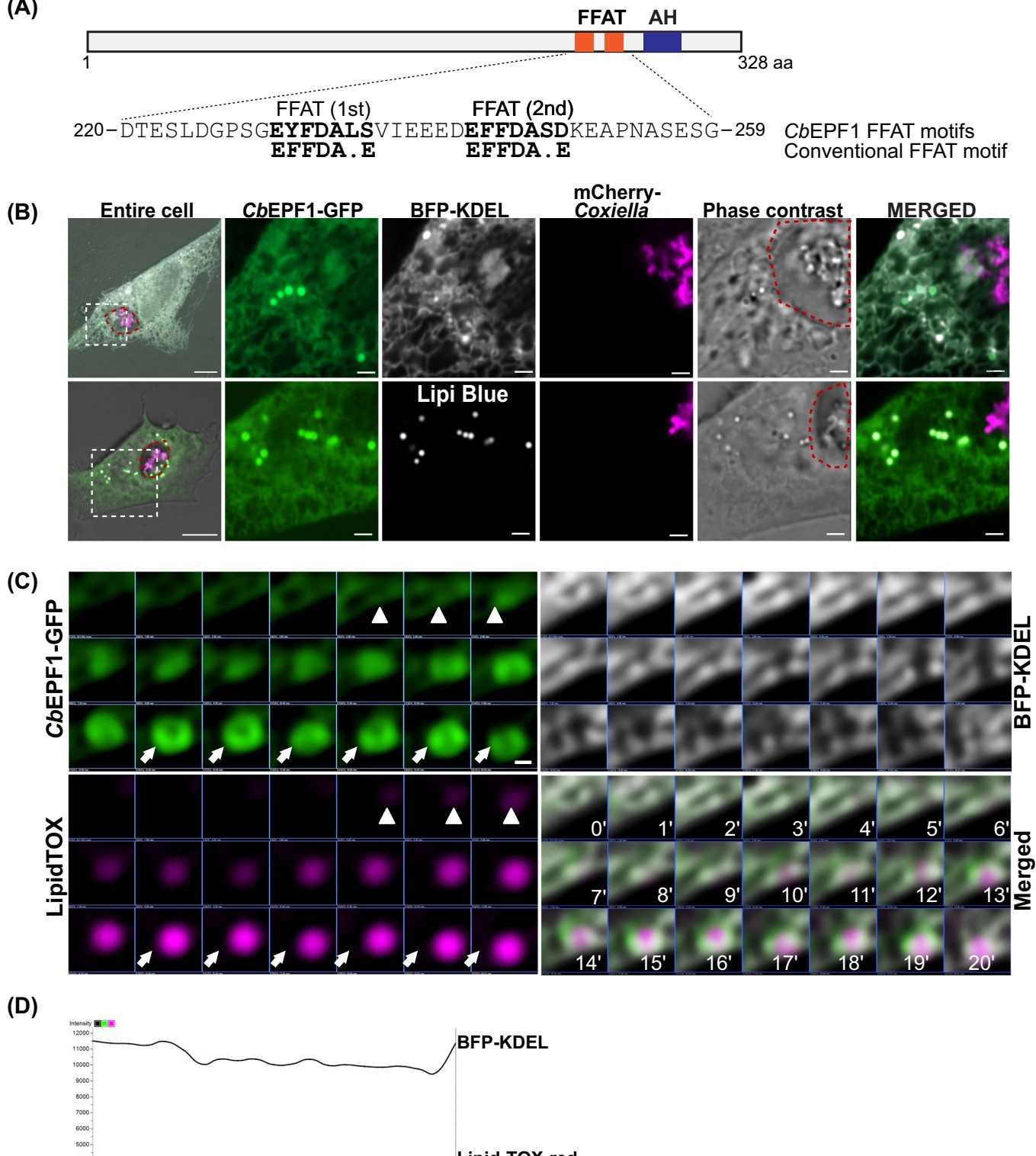

**Figure 1. CbEPF1-GFP localizes to host ER and LDs.**

(A) Two putative FFAT motifs were identified in the C-terminal region of CbEPF1 protein. The sequence and position of the FFAT motifs are shown, with amino acid numbers indicated. Conventional FFAT motif sequence is shown below predicted CbEPF1 FFAT motifs. AH represents a predicted site of amphipathic helix. (B) Live cell microscopy shows that ectopically expressed CbEPF1-GFP localized to the host ER and LDs in mCherry-Coxiella infected HeLa cells. The ER was labeled with BFP-KDEL and LDs were labeled with LipidTOX-red. The Coxiella-containing vacuole (CCV) membrane is outlined in red in the phase image. Scale bar 10 μm (overview) and 2 μm (magnified). (C) CbEPF1-GFP associated with LD biogenesis in the ER. Arrowheads indicate early-stage LD biogenesis sites ($t = 4$–6 min) and arrows mark larger LDs with CbEPF1-GFP localized on the entire LD surface ($t > 13$ min). Images were acquired every 1 min ($t = 0$–20 min) by spinning disc confocal microscopy. Scale bar 0.2 μm. (D) Intensity profile of BFP-KDEL, CbEPF1-GFP, and LipidTOX-red from Fig. 1C during LD biogenesis in the ER from $t = 0$–20 min. From 4 min time point, the increase in LipidTOX-red intensity indicates emergence and growth of LD.

CbEPF1–VAPB interactions in vitro, we cannot exclude the association/involvement of other proteins at the CbEPF1–VAPB molecular tether.

## CbEPF1 FFAT motifs show preferential interaction among VAP family proteins

We next sought to validate CbEPF1 interaction with VAPs in mammalian cells. First, we ectopically expressed GFP or CbEPF1-GFP in HeLa cells and examined co-localization of endogenous VAPs (VAPA and VAPB) with respect to CbEPF1 and LDs after oleate treatment. In GFP-expressing cells, VAPA and VAPB exhibited a reticulate pattern and minimal interaction with LDs (Fig. 3D, top panel; Fig. 3E, top panel), confirming previous reports that VAPA and VAPB do not associate with LDs (Zouiouich et al, 2022). In contrast, cells expressing CbEPF1-GFP showed VAPA and VAPB forming ring-like structures around CbEPF1-GFP-positive LDs (Fig. 3D, bottom panel; Fig. 3E, bottom panel). Due to the inability to detect endogenous MOSPD2 using available antibodies, we ectopically expressed GFP-MOSPD2 and mCherry-CbEPF1 in HeLa cells to determine MOSPD2 localization in relation to CbEPF1 and LDs. As previously reported, MOSPD2 localized around LDs irrespective of CbEPF1 expression (Fig. 3F) (Zouiouich et al, 2022). Hence, we hypothesize that VAPA and VAPB localization around CbEPF1-GFP positive LDs is due to CbEPF1-GFP.

To further evaluate CbEPF1 interaction with VAPs in mammalian cells, we expressed CbEPF1-GFP or its FFAT mutant variants in HEK293T cells and induced LD formation with OA prior to immunoprecipitation. In western blot analysis, VAPA and VAPB co-immunoprecipitated with CbEPF1-GFP, CbEPF1-F1mt-GFP, and CbEPF1-F2mt-GFP but not with CbEPF1-F3mt-GFP or GFP (Fig. 3G,H). This suggests VAPA and VAPB interacts with both CbEPF1 FFAT motifs. Interestingly, another VAP family protein, MOSPD2, co-immunoprecipitated with CbEPF1-GFP and CbEPF1-F1mt-GFP but not with CbEPF1-F2mt-GFP and CbEPF1-F3mt-GFP. This suggests that unlike VAPA and VAPB, MOSPD2 interacts with only the second CbEPF1 FFAT motif (Fig. 3G). While VAPA, VAPB, and MOSPD2 share similar MSP domain structure and affinities for consensus FFAT motifs (Di Mattia et al, 2018; Di Mattia et al, 2020), our observation that MOSPD2 exhibits preferential binding to the second CbEPF1 FFAT motif (EFF-DASD) over the first FFAT motif (EYFDALS) provides additional evidence that MOSPD2 and VAP-A/B do not always bind the same FFAT motifs (Cabukusta et al, 2020). In conclusion, our data demonstrate that the two FFAT motifs in CbEPF1 display preferential interaction for VAP family proteins.

## CbEPF1 FFAT motifs mediate ER-LD contacts

As we demonstrated that the two CbEPF1 FFAT motifs and the VAP MSP domain are responsible for CbEPF1–VAP interaction, we next analyzed whether the CbEPF1 FFAT motifs mediate ER-LD contact sites. Using ectopic expression of GFP fusion proteins, we studied the role of each FFAT motif in CbEPF1-mediated ER-LD contact sites, based on association of the ER with LDs. Mutations in the CbEPF1 FFAT motif(s) did not influence CbEPF1-GFP localization to either the host ER or LDs (Fig. EV2). However, only CbEPF1-F1mt-GFP and CbEPF1-F2mt-GFP proteins induce ER-LD contacts similar to CbEPF1-Wt, whereas CbEPF1-F3mt-GFP failed to induce ER-LD contacts (Fig. 4A,B). These results suggest that at least one functional FFAT motif is necessary and sufficient for CbEPF1 to mediate ER-LD contact sites and rearrange the ER around LDs. Interestingly, CbEPF1-F3mt-GFP expression in cells led to LD clustering, as defined by closer proximity, and exclusion of the ER in the vicinity of LD clusters, which are random in size (Figs. 4B,C and EV3). Since the CbEPF1 FFAT motifs are essential for interaction with VAP proteins in the ER, the absence of functional FFAT motifs in CbEPF1-F3mt-GFP disrupts the ER-LD interaction. As a result, this may increase LD-LD interactions through unidentified LD surface proteins, leading to the clustered phenotype.

We next tested whether ER-LD contacts exist in C. burnetii-infected cells. We previously demonstrated that C. burnetii T4BSS effector proteins induce an increased number of LDs in macrophages (Mulye et al, 2018). Therefore, we measured the extent of LD-ER contacts (<40 nm distance) in uninfected and C. burnetii-infected macrophages using transmission electron microscopy (TEM). C. burnetii-infected macrophages showed significantly higher ER-LD contacts compared to control cells (Fig. 5A,B). Taken together, these findings suggest that C. burnetii infection, through the secreted effector protein CbEPF1, increases ER-LD contacts in host cells.

## An amphipathic helix (AH) and a hydrophobic region target CbEPF1 to LDs

Cytosolic proteins which localize to LDs typically contain amphipathic helix (AH) regions that recognize phospholipid packing defects at the LD monolayer surface (Prévost et al, 2018). In contrast, ER proteins localize to LDs through hydrophobic, membrane-embedded motifs that acquire a hairpin conformation in the endoplasmic reticulum (Olarte et al, 2022; Olarte et al, 2020). To identify how CbEPF1 localizes to LDs, we analyzed the predicted secondary structure using AmphipaSeek (Sapay et al, 2006) and HeliQuest (Gautier et al, 2008) and identified a putative

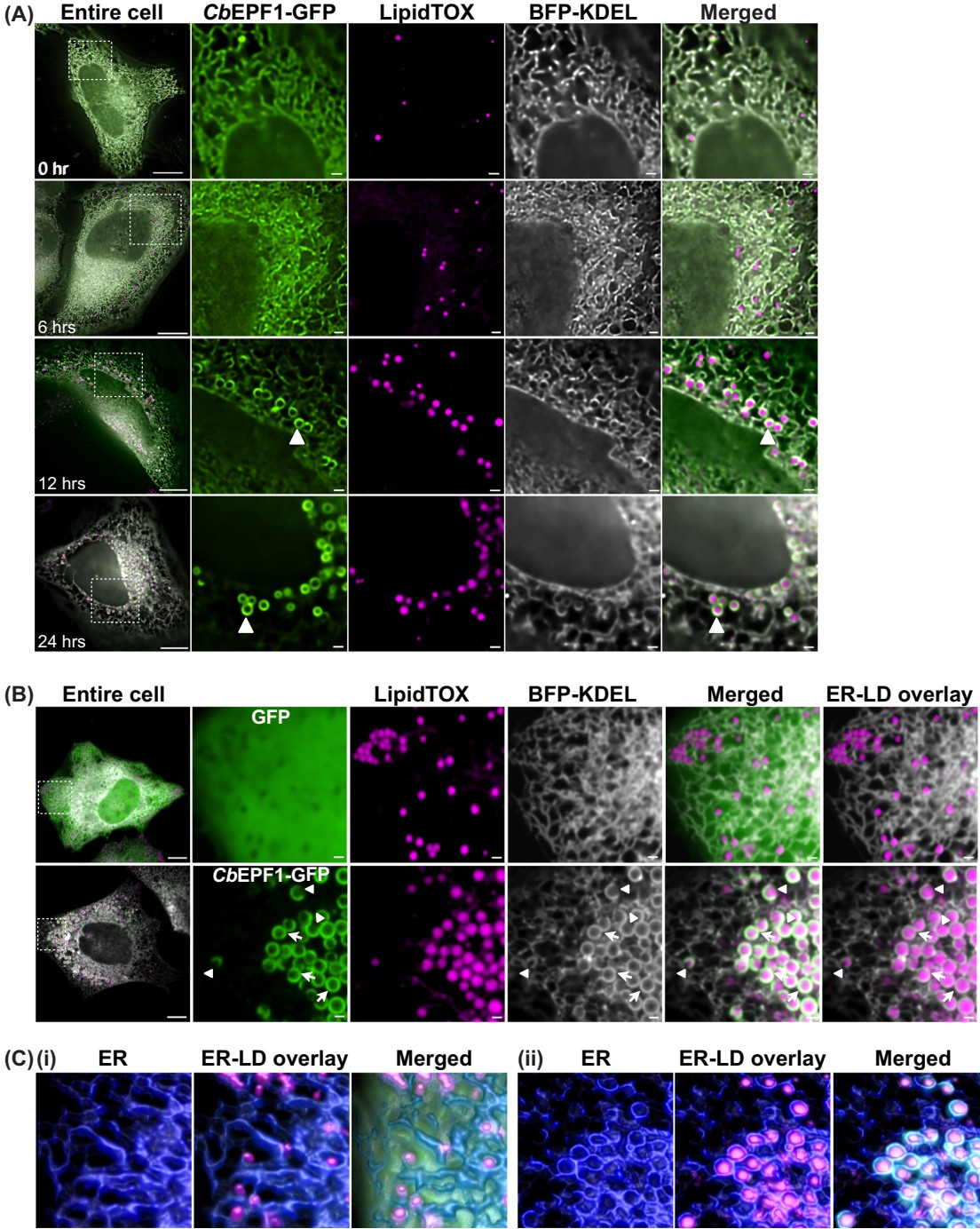

**Figure 2. CbEPF1 associates with host LDs and induces inter-organelle contact sites between the host ER and LDs.**

(A) HeLa cells expressing CbEPF1-GFP and BFP-KDEL were treated with OA (30 μM) and imaged every 6 h by live cell spinning disc microscopy. Representative deconvoluted images show CbEPF1-GFP relocates from the ER to the LD surface as LDs grow (arrowhead). Scale bar 10 μm (overview) and 1 μm (magnified). (B) HeLa cells expressing GFP or CbEPF1-GFP and BFP-KDEL were treated with OA (100 μM, overnight) to induce LD biogenesis. LDs were visualized with LipidTOX red. ER shows limited association with LDs in GFP expressing cells (top panel). CbEPF1-GFP expression induces extended contacts between ER and LDs (bottom panel). Arrows show ER wrapping around CbEPF1 localized LDs. Arrowheads show absence of ER interaction at the regions of LDs where CbEPF1-GFP localization is absent. Scale bar 10 μm (overview) and 1 μm (magnified). (C) Representative image of 3D rendering of a Z-stack image illustrating the association between LDs (magenta) and ER (blue). (i) In cells expressing GFP alone, the ER exhibits minor contacts with LDs. (ii) In contrast, cells expressing CbEPF1-GFP display extended ER-LD contacts, with ER-LD interactions specifically localized to regions where CbEPF1-GFP (green) is present on LDs.

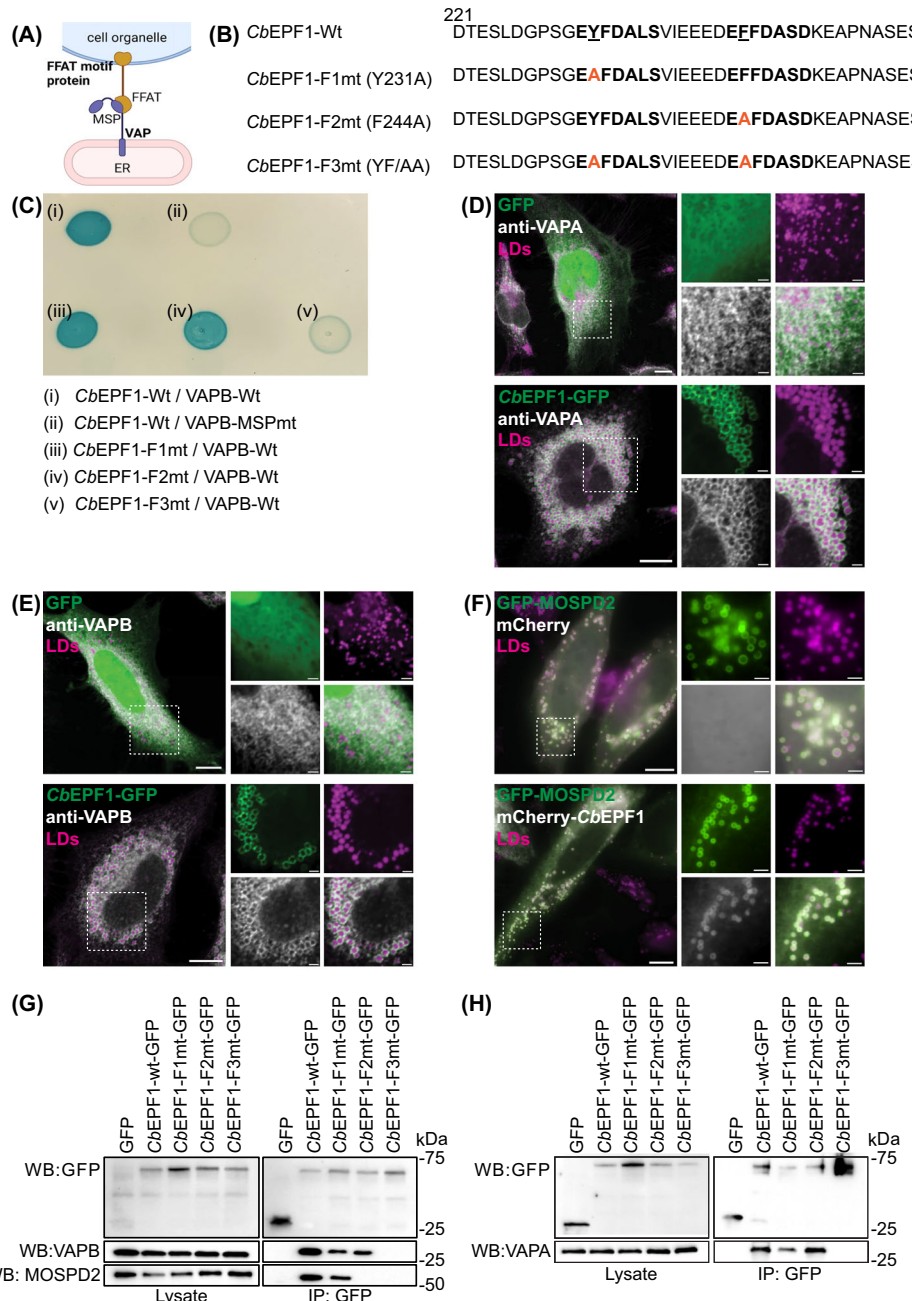

**Figure 3. *Cb*EPF1 FFAT motifs interact with the VAP MSP domain.**

(A) Schematic representation of the FFAT motif-containing protein interaction with the ER protein VAP to form inter-organelle contact sites between ER and LDs. Created with BioRender.com. (B) *Cb*EPF1 amino acid sequence contains two FFAT motifs. Number indicates amino acid position; essential position (2nd residue) in FFAT motif is underlined. Mutations in FFAT motif(s) are highlighted in orange. (C) Bacterial adenylate cyclase-based two-hybrid assay (BACTH) shows *Cb*EPF1 interacts with VAPB in an MSP-FFAT motif-dependent manner. (D) HeLa cells expressing GFP (green) or *Cb*EPF1-GFP (green) were treated with OA (100 μM, overnight) and labeled with anti-VAPA (white). LDs were stained with Lipi Blue (magenta). Scale bars: 10 μm (overview) and 2 μm (magnified). (E) HeLa cells expressing GFP (green) or *Cb*EPF1-GFP (green) were treated with OA (100 μM, overnight) and labeled with anti-VAPB (white). LDs were stained with Lipi Blue (magenta). Scale bars: 10 μm (overview) and 2 μm (magnified). (F) HeLa cells expressing GFP-MOSPD2 (green) along with mCherry (white) or mCherry-*Cb*EPF1 (white) were treated with OA (100 μM, overnight) and LDs were stained with Lipi Blue (magenta). Scale bars: 10 μm (overview) and 2 μm (magnified). (G, H) Immunoprecipitation of GFP and *Cb*EPF1-GFP (WT, F1mt, F2mt, and F3mt) from lysates of HEK293 induced with OA (100 μM, overnight). WB represents western blot analysis using respective primary antibody. Source data are available online for this figure.

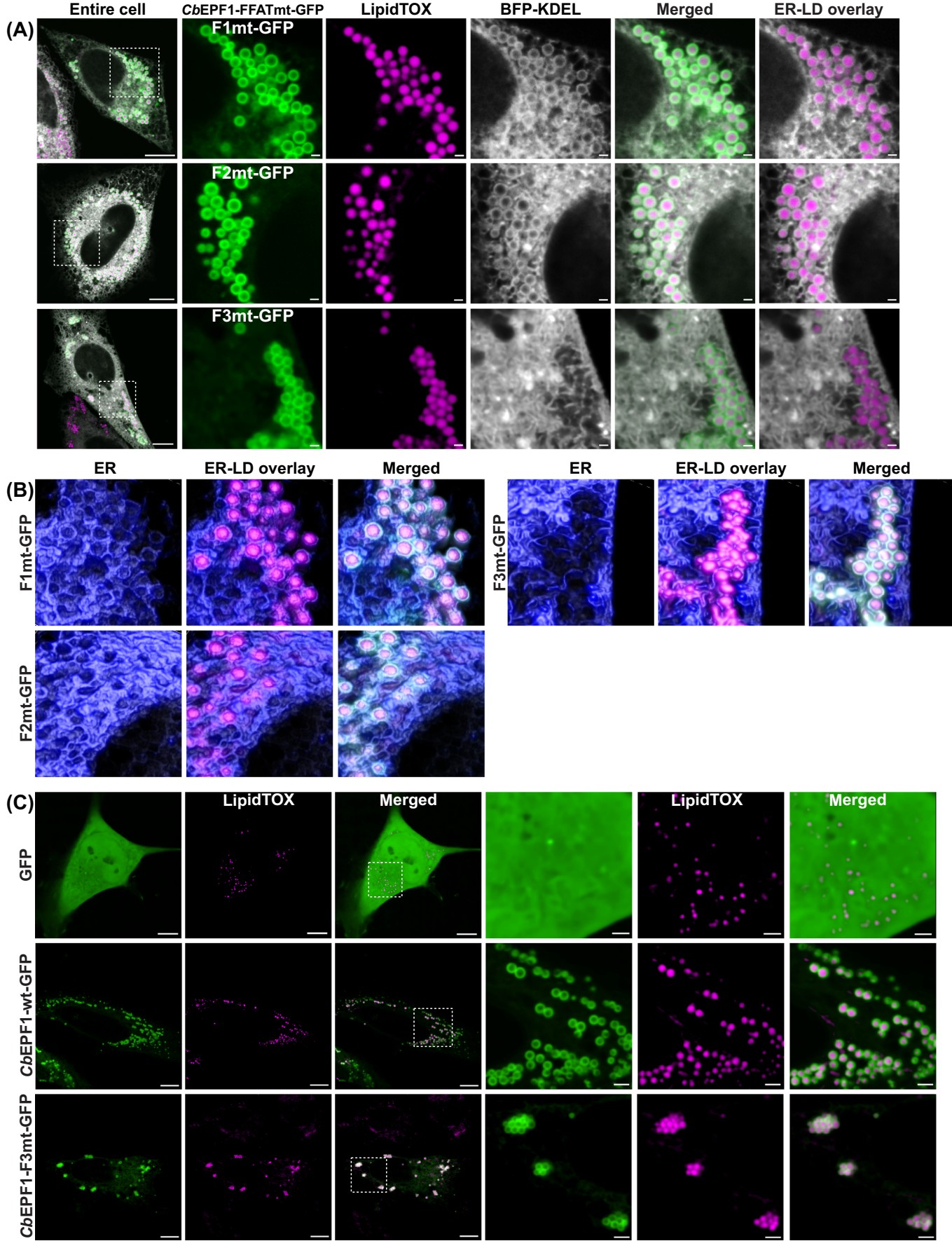

**Figure 4. Functional FFAT motifs are required for *Cb*EPF1-induced ER-LDs contact sites.**

(A) *Cb*EPF1-F1mt-GFP and *Cb*EPF1-F2mt-GFP induce ER-LD contact sites while *Cb*EPF1-F3mt-GFP failed to induce ER-LD contact sites. Scale bar 10 μm (overview) and 1 μm (magnified). (B) Representative image of 3D rendering of a Z-stack image illustrating the association between LDs (magenta) and ER (blue) in *Cb*EPF1-FFAT mutant-GFP expressing cells. *Cb*EPF1-F1mt-GFP or *Cb*EPF1-F2mt-GFP expressing cells exhibited ER-LD contacts, while the *Cb*EPF1-F3mt-GFP expressing cells showed exclusion of ER around LD clusters. (C) *Cb*EPF1-F3mt-GFP expression in cells causes clustering of LDs. Scale bar 10 μm (overview) and 1 μm (magnified).

amphipathic helix (AH) in the C-terminal region of *Cb*EPF1 (amino acids 262–296) (Fig. 6A). The AH was disrupted by replacing the hydrophobic residues in the middle of the nonpolar face with positively charged arginine residues (*Cb*EPF1-W270R/L271R/L274R/L275R, referred to as *Cb*EPF1-AHmt) (Fig. 6A) and expressed in oleate-treated HeLa cells. Compared to *Cb*EPF1-WT-GFP, which localized to the entire LD population (100%) in oleate-treated HeLa cells (Fig. 6B), *Cb*EPF1-AHmt-GFP localized to a subpopulation (40%) of LDs (Fig. 6C). To further examine whether the AH region is sufficient for LD localization, and whether the disrupting the AH is sufficient to block LD localization, we ectopically expressed the AH and AHmt regions as GFP fusions. In oleate-treated HeLa cells, AH-GFP localized to LDs (Fig. 6D) and AHmt-GFP showed diffused localization (Fig. 6E), indicating that the identified AH region localizes to LDs, and the AH mutations are sufficient to disrupt LD localization.

Because the AH mutations were not sufficient to completely disrupt *Cb*EPF1-GFP localization to LDs, we hypothesized that additional motifs may be involved. To test this, a series of *Cb*EPF1 truncation mutants tested for localization in oleate-treated HeLa cells. While the N-terminal 101 amino acids (*Cb*EPF1-N(1-101)-GFP) did not show specific localization (Fig. 6F), the middle 111 amino acids of *Cb*EPF1 (*Cb*EPF1-M(101-212)-GFP) showed specific localization to LDs (Fig. 6G). This suggests that, in addition to the amphipathic helix in the C-terminal region, *Cb*EPF1 contains additional LD localizing signal(s) in the middle 111 amino acids, a region which is rich in hydrophobic amino acids.

Next, we tested whether the C-terminal region with AH alone and lacking the middle hydrophobic region, was sufficient for *Cb*EPF1 localization to LDs. We observed that the *Cb*EPF1-C(213-328)-GFP localized to LDs (Fig. 6H) and that mutations in the AH region disrupted *Cb*EPF1-C-AHmt(213-328)-GFP localization to LDs (Fig. 6I). This suggests that the *Cb*EPF1 AH region is sufficient for localization to LDs. Finally, we tested whether *Cb*EPF1 can localize to LDs without the C-terminus, including the AH (*Cb*EPF1(1-252)-GFP). Interestingly, *Cb*EPF1(1-252)-GFP, although containing the middle hydrophobic region, failed to show specific localization (Fig. 6J). This suggests that while either domain (the middle hydrophobic region or the amphipathic helix) are sufficient to localize *Cb*EPF1 to LDs, both are likely crucial for the structural conformation or stabilizing *Cb*EPF1 on LDs.

### *Cb*EPF1 regulates host LD metabolism in a FFAT-dependent manner

Next, we sought to evaluate whether *Cb*EPF1 impacts host LD metabolism by measuring LD numbers in cells expressing *Cb*EPF1-GFP or the *Cb*EPF1-GFP FFAT mutants. Compared to cells expressing GFP alone, cells expressing *Cb*EPF1-GFP or *Cb*EPF1-FFAT mutant GFP showed significantly high number of LDs (Fig. 7A), suggesting *Cb*EPF1 expression impacts LD number independent of the FFAT motif.

As *Cb*EPF1 induces ER-LD contacts, we sought to evaluate whether there is an impact on LD size upon OA treatment. We supplemented cells expressing *Cb*EPF1-GFP or *Cb*EPF1-GFP FFAT mutants with OA and measured the LD diameter. *Cb*EPF1-GFP expressing cells showed significantly larger LDs compared to cells expressing GFP. Expression of the single FFAT mutants, *Cb*EPF1-F1mt-GFP and *Cb*EPF1-F2mt-GFP, resulted in larger LDs compared to control GFP-expressing cells, but LD sizes relatively similar to *Cb*EPF1-GFP expressing cells. However, cells expressing the double FFAT mutant *Cb*EPF1-F3mt-GFP contained LDs similar in size to GFP control cells and significantly smaller LDs compared to *Cb*EPF1-GFP or *Cb*EPF1-F1mt-GFP or *Cb*EPF1-F2mt-GFP expressing cells (Fig. 7B,C). These results suggest that *Cb*EPF1 promotes the development of larger LDs in its FFAT motif-dependent manner.

## Discussion

Inter-organelle contact sites in eukaryotic cells serve as pivotal hubs for cellular homeostasis. Recent research has unveiled the strategies employed by pathogens to target these sites and regulate host cell metabolism (Dumoux and Hayward, 2016; Derré, 2017; Jiang et al, 2022). While viruses have been recognized for their manipulation of host inter-organelle contact sites, bacterial-mediated inter-organelle contact sites were previously limited to the ER and the BCV (Justis et al, 2017; Stoeck et al, 2018; Derré et al, 2011; Cook et al, 2022; Yue et al, 2023). Our study unveils the first example of a secreted bacterial effector protein, *Cb*EPF1, that orchestrates inter-organelle contact sites beyond the confines of the BCV, thereby exerting control over host cellular lipid metabolism. These results underscore that the FFAT motifs in *Cb*EPF1 play a central role in mediating inter-organelle contact sites between the host ER and LDs by interacting with VAPs in the host ER. Furthermore, these contact sites influence the growth of LDs, a critical aspect of lipid metabolism.

Our data support a model where *Cb*EPF1 translocates from the ER to LDs at sites of LD formation. While LD-associated *Cb*EPF1 FFAT motifs interact with VAPs on the ER, it is unclear whether ER-localized *Cb*EPF1 also interacts with VAPs. Given that the ER is characterized by a double-membrane structure distinct from the monolayer structure of the LD, we hypothesize that *Cb*EPF1 may undergo conformational changes as it moves from the ER to the LD surface. Consequently, *Cb*EPF1 could potentially adopt a structure that allows it to interact with VAPs only when localized on LDs. While our study is limited technically due to a lack of a *Cb*EPF1-specific antibody or detection of *C. burnetii*-secreted *Cb*EPF1, future studies elucidating *Cb*EPF1 localization and structure, both spatially and temporally, during LD biogenesis may reveal a novel bacterial strategy to manipulate the host cell.

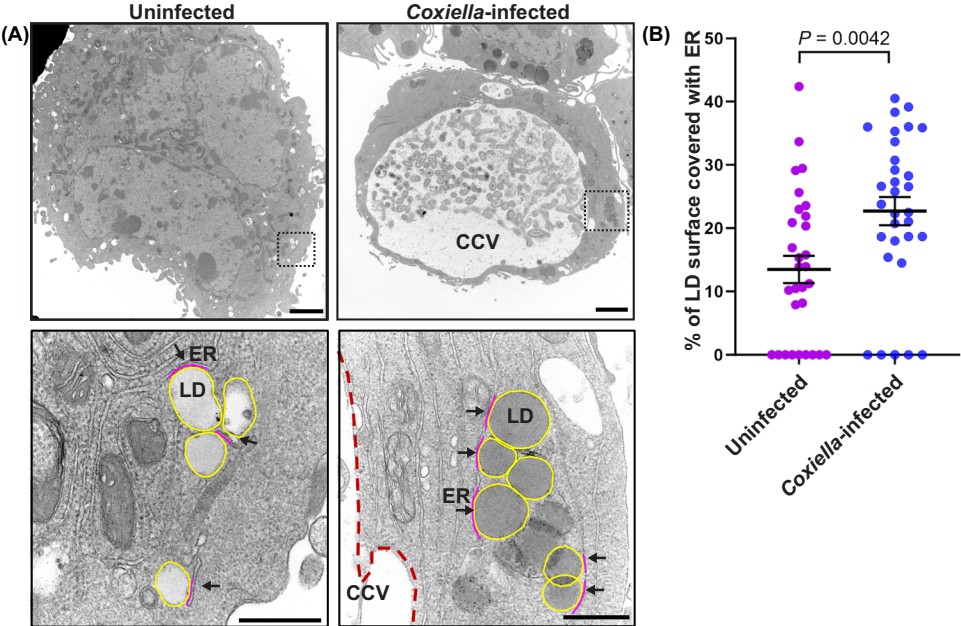

**Figure 5. *C. burnetii* infection induces ER-LD contacts in macrophages.**

(A) Representative transmission electron micrograph of uninfected or *C. burnetii*-infected MH-S cells. Black arrows mark the ER-LD contact areas that are less then 40 nm distance. Yellow lines are LDs, Magenta lines are endoplasmic reticulum and red dashed lines are CCV membrane. CCV, *C. burnetii* Containing Vacuole; LD, lipid droplets; ER, endoplasmic reticulum. Scale bar represents 2 μm (overview) and 500 nm (magnified). (B) Quantification of LD surface covered with ER at <40 nm distance. Random LDs from ten different cells were measured to calculate % of LD surface covered with ER ($n = 30$ LDs for uninfected cells and $n = 31$ LDs for *Coxiella*-infected cells). Data was collected from a single experiment. Statistical analysis was performed using Two-tailed Unpaired Student's *t*-test. $P = 0.0042$ (Uninfected vs *Coxiella*-infected). Error bars represent the mean ± SEM. Source data are available online for this figure.

In the absence of FFAT motifs, *Cb*EPF1-F3-mt induces LD clustering without affecting LD size. The physiological importance of the LD clustering phenomenon is unclear but has been suggested to precede LD fusion. For example, the LD-associated proteins FSP27 (CIDEC) and GRAF1a induce LD clustering and promote LD fusion to generate larger LDs (Jambunathan et al, 2011; Lucken-Ardjomande Häsler et al, 2014). However, depleting the LD-associated proteins Seipin, Atg 2A, and Atg 2B results in the clustering of irregularly sized LDs (Salo et al, 2016; Szymanski et al, 2007; Velikkakath et al, 2012). Interestingly, other proteins such as Rab40c, DFCP1, AUP1, CG9186, and HCV core proteins induce LD clustering without causing larger LDs (Tan et al, 2013; Li et al, 2019; Lohmann et al, 2013; Thiel et al, 2013; Depla et al, 2010). While proteins may mediate LD-LD interactions, lipids play a key role in LD fusion, with higher levels of phosphatidic acid (PA) levels promoting LD fusion, whereas higher levels of phosphatidylcholine (PC) or squalene inhibit LD fusion (Fei et al, 2011; Krahmer et al, 2011; Ta et al, 2012). These findings suggest that the balance of specific lipids, regulated by LD-associated proteins, plays a crucial role in lipid storage and metabolism. It is possible that *Cb*EPF1 may influence the LD surface properties, either directly or indirectly through unidentified protein-protein interactions, which further affect lipid metabolism.

*Cb*EPF1 localization to LDs is directed by both a middle hydrophobic domain and a C-terminal amphipathic helix region. However, it is unknown how these two domains function together to target and/or stabilize *Cb*EPF1 on LDs. Proteins encoding two separate domains to control LD localization are not uncommon.

Hydroxysteroid 17-beta-dehydrogenase 13 (HSD17B13), a hepatic lipid droplet-associated enzyme, contains an N-terminal hydrophobic helix and a C-terminal amphipathic helix which target the protein to LDs (Liu et al, 2023). Based on the structural information of the HSD17B13 dimer, the proposed model for LD targeting suggests that the N-terminal hydrophobic helices embed in the lipid membrane, while the amphipathic helices position the hydrophobic face into the membrane's interior such that the charged face interacts with lipid head groups (Liu et al, 2023). Additional experiments, including determining *Cb*EPF1 structure, are needed to fully elucidate how *Cb*EPF1 interacts with and binds to LDs.

Several bacterial effector proteins localize to host LDs or are involved in translocating cytoplasmic LDs to BCVs (Kumar et al, 2006; Cocchiaro et al, 2008; Bugalhão et al, 2022; Chen et al, 2022). While *Cb*EPF1 does not share sequence or functional homology with identified LD-associated bacterial effector proteins, our findings do suggest that *Cb*EPF1 exhibits functional homology with eukaryotic proteins known to relocalize from ER to LD surface. For example, the eukaryotic proteins GPAT4, DFCP1, LDAF1, and Rab18 relocalize from ER to LD surface, ultimately to promote LD growth (Wilfling et al, 2013; Xu et al, 2018; Chung et al, 2019; Li et al, 2019). This functional similarity underscores the concept that bacterial effectors possess either undiscovered domains or eukaryotic-like short linear motifs (SLiMs) as a mechanism to mimic host proteins and exert control over host cellular pathways. Significantly, mounting evidence supports the utilization of eukaryotic-like SLiMs by bacterial effectors as a

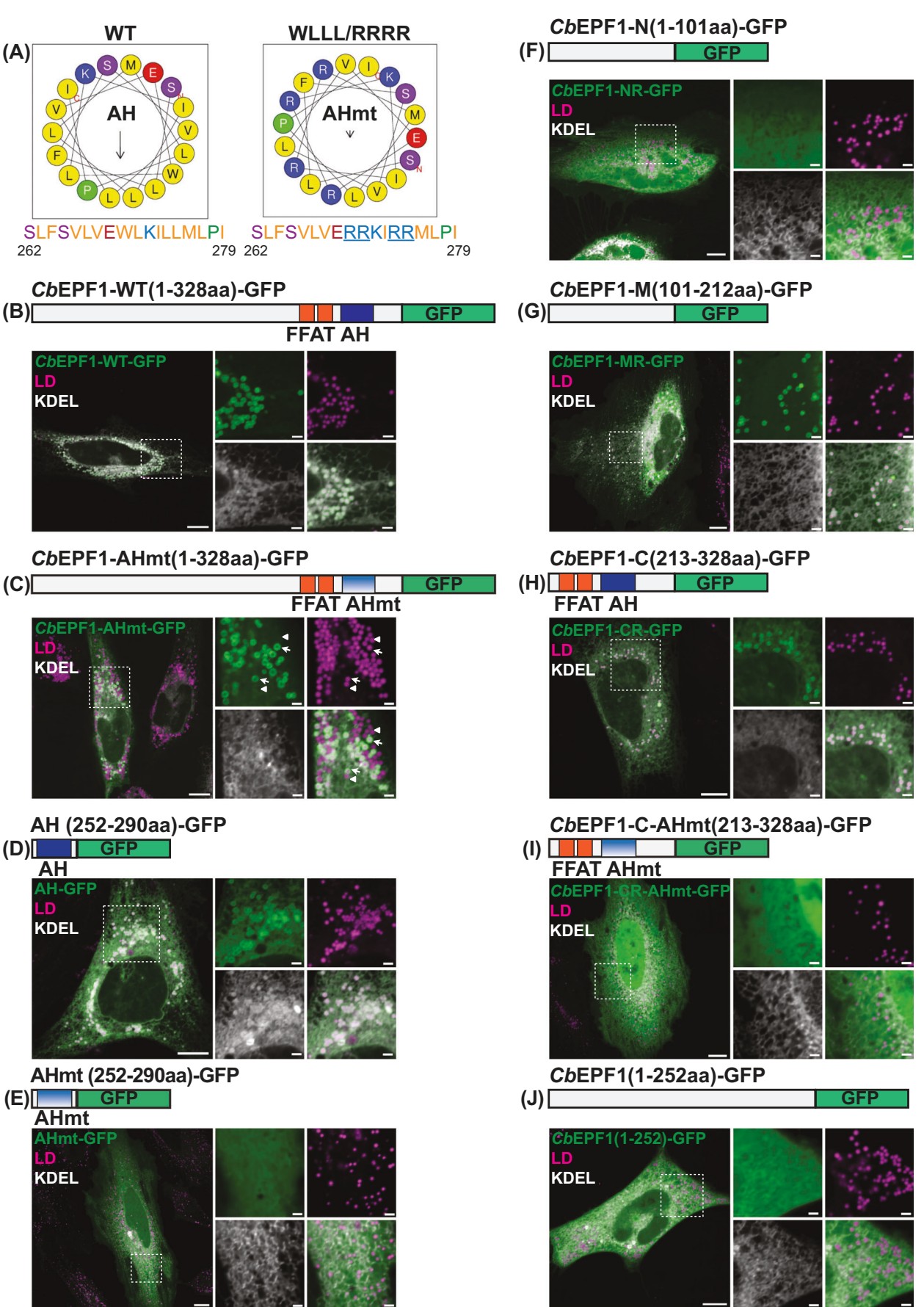

**Figure 6.  An amphipathic helix (AH) in the C-terminal region and a hydrophobic region in the middle of *Cb*EPF1 are crucial for LD targeting.**

(A) Helical wheel representation of the wild-type amphipathic helix (AH) (left) and mutant amphipathic helix (AHmt) (right) (aa 262–279) generated with HeliQuest. The mutations in AHmt alters the amphipathic character of the helix by reducing its hydrophobic moment ($\mu$H) from 0.403 (AH) to 0.103 (AHmt). (B) Top: Schematic representation of full-length *Cb*EPF1-WT as GFP fusion. Bottom: Localization of the *Cb*EPF1-WT-GFP (green) along with BFP-KDEL (white) in HeLa cells treated with OA; LDs were stained with Lipi Red (magenta). Scale bars: 10 μm (overview) and 2 μm (magnified). (C) Top: Schematic representation of full-length *Cb*EPF1-WT with mutations in AH region as GFP fusion. Bottom: Localization of the *Cb*EPF1-AHmt-GFP along with BFP-KDEL (white) in HeLa cells treated with OA; LDs were stained with Lipi Red (magenta). Arrow heads show LDs without *Cb*EPF1-AHmt-GFP localization and arrows show LDs with *Cb*EPF1-AHmt-GFP localization. Scale bars: 10 μm (overview) and 2 μm (magnified). (D) Top: Schematic representation of WT amphipathic helix region (AH) as GFP fusion. Bottom: Localization of the AH-GFP along with BFP-KDEL (white) in HeLa cells treated with OA; LDs were stained with Lipi Red (magenta). Scale bars: 10 μm (overview) and 2 μm (magnified). (E) Top: Schematic representation of mutant amphipathic helix region (AHmt) as GFP fusion. Bottom: Localization of the AHmt-GFP along with BFP-KDEL (white) in HeLa cells treated with OA; LDs were stained with Lipi Red (magenta). Scale bars: 10 μm (overview) and 2 μm (magnified). (F) Top: Schematic representation of N-terminal 101 amino acids of *Cb*EPF1 (*Cb*EPF1-N(1-101aa)) as GFP fusion. Bottom: Localization of the *Cb*EPF1-N(1-101aa)-GFP along with BFP-KDEL (white) in HeLa cells treated with OA; LDs were stained with Lipi Red (magenta). Scale bars: 10 μm (overview) and 2 μm (magnified). (G) Top: Schematic representation of middle 111 amino acids of *Cb*EPF1 (*Cb*EPF1-M(101-212aa)) as GFP fusion. Bottom: Localization of the *Cb*EPF1-M(101-212aa)-GFP along with BFP-KDEL (white) in HeLa cells treated with OA; LDs were stained with Lipi Red (magenta). Scale bars: 10 μm (overview) and 2 μm (magnified). (H) Top: Schematic representation of C-terminal 115 amino acids of *Cb*EPF1 (*Cb*EPF1-C(213-328aa)) as GFP fusion. Bottom: Localization of the *Cb*EPF1-C(213-328aa)-GFP along with BFP-KDEL (white) in HeLa cells treated with OA; LDs were stained with Lipi Red (magenta). Scale bars: 10 μm (overview) and 2 μm (magnified). (I) Top: Schematic representation of C-terminal 115 amino acids of *Cb*EPF1 with mutations in AH region (*Cb*EPF1-C-AHmt(213-328aa)) as GFP fusion. Bottom: Localization of the *Cb*EPF1-C-AHmt(213-328aa)-GFP along with BFP-KDEL (white) in HeLa cells treated with OA; LDs were stained with Lipi Red (magenta). Scale bars: 10 μm (overview) and 2 μm (magnified). (J) Top: Schematic representation of N-terminal 252 amino acids of *Cb*EPF1 (*Cb*EPF1(1-252aa)) as GFP fusion. Bottom: Localization of the *Cb*EPF1(1-252aa)-GFP along with BFP-KDEL (white) in HeLa cells treated with OA; LDs were stained with Lipi Red (magenta). Scale bars: 10 μm (overview) and 2 μm (magnified).

strategy to hijack cellular machinery (Via et al, 2015; Sámano-Sánchez and Gibson, 2020). The *Cb*EPF1 FFAT motifs serve as a compelling example of eukaryotic-like SLiMs in bacterial proteins and extends our understanding of these motifs and their adaptability in pathogenic contexts. The intriguing functional homology between *Cb*EPF1 and eukaryotic proteins also raises questions about the precise mechanisms through which *Cb*EPF1 modulates host lipid metabolism. Although the exact structure of the N-terminal region of *Cb*EPF1 is unknown, based on the predicted structure available in AlphaFold2 Protein Structure Database (Varadi et al, 2024), the *Cb*EPF1 N-terminal region may form a globular domain composed of several helices (UniProt: Q83BW7). This hypothetical globular domain may function at the ER-LD interface, possibly in transferring neutral lipids or sterol esters to LDs. Further structural and biochemical studies are needed to confirm the presence of a globular domain(s) within the N-terminal region of *Cb*EPF1 and to understand its function. While our findings suggest a role for *Cb*EPF1 in lipid transport at ER-LD contacts, whether *Cb*EPF1 functions directly as a lipid transporter or as a molecular tether that recruits additional lipid transporters to the ER-LD contact sites has yet to be determined. The presence of two FFAT motifs in *Cb*EPF1 is an intriguing evolutionary adaptation that potentially enhances its interaction with VAPs in the ER, expanding the ER-LD contact area. Furthermore, the preferential interaction of the second *Cb*EPF1 FFAT motif with MOSPD2, a sole VAP that is known to localize on LD surfaces (Zouiouich et al, 2022), raises a possible complex functional role of *Cb*EPF1 on host LDs metabolism. *Cb*EPF1 underscores the pathogen's capacity to manipulate host cell inter-organelle communication, emphasizing the need for further exploration of FFAT motif-mediated inter-organelle contacts in the context of pathogen infections.

During infection of macrophages, *C. burnetii* T4BSS secreted effector protein(s) induce lipid droplet accumulation (Mulye et al, 2018). However, the specific secreted effector protein and its molecular mechanism remain unknown. *Cb*EPF1, which initially localizes to LD biogenesis sites in the ER and later translocates to LD surfaces, plays an active role in LD formation and growth.

Based on these findings, we propose that *Cb*EPF1 is a T4BSS effector protein responsible for lipid droplet accumulation during *C. burnetii* infection. The specific benefits of LDs for *C. burnetii* survival or replication are not fully understood. Our previous study on the impact of host LD homeostasis on *C. burnetii* infection revealed that LD breakdown supports *C. burnetii* growth (Mulye et al, 2018). Therefore, we speculate that the lipids and sterols derived from accumulated host LDs might serve as an energy source or produce lipid immune modulators that benefit *C. burnetii*. An alternate benefit of LD accumulation could be as a protective mechanism against cholesterol toxicity to *C. burnetii*. *C. burnetii* survives and replicates within a specialized *Coxiella*-containing vacuole (CCV) that is rich in sterols (Howe and Heinzen, 2006). However, our previous research demonstrated that cholesterol accumulation in the CCV membrane is detrimental to *C. burnetii* growth (Mulye et al, 2017). Consequently, *C. burnetii* has evolved multiple molecular strategies to modulate CCV membrane cholesterol levels, thereby avoiding toxicity (Justis et al, 2017; Clemente et al, 2022; Schuler et al, 2023). Notably, we previously described how *C. burnetii* hijacks the host sterol transporter ORP1L, which upon recruitment to the CCV membrane interacts with VAPs in the ER to establish inter-organelle contact sites between the ER and CCV. This interaction facilitates cholesterol efflux from the CCV to the ER (Justis et al, 2017; Schuler et al, 2023). In this study we demonstrate that *C. burnetii* infection increases ER-LD contacts in the host cells. Overall, our findings suggest a possible mechanism for *C. burnetii* T4BSS effector proteins to redirect cholesterol away from the ER towards storage in LDs. Given that our present experiments relied on ectopic expression, future studies will benefit from the use of *C. burnetii* *Cb*EPF1 mutants as additional tools become available.

In conclusion, our results illuminate the multifaceted role of *Cb*EPF1 as a regulator of host LDs by intervening in host inter-organelle communication. Serving as a tether protein between the host ER and LDs through its FFAT motifs, *Cb*EPF1 showcases the ability of a bacterial effector protein to manipulate inter-organelle communication beyond BCVs. As we continue to delve into the intricacies of this interplay, future research should focus on

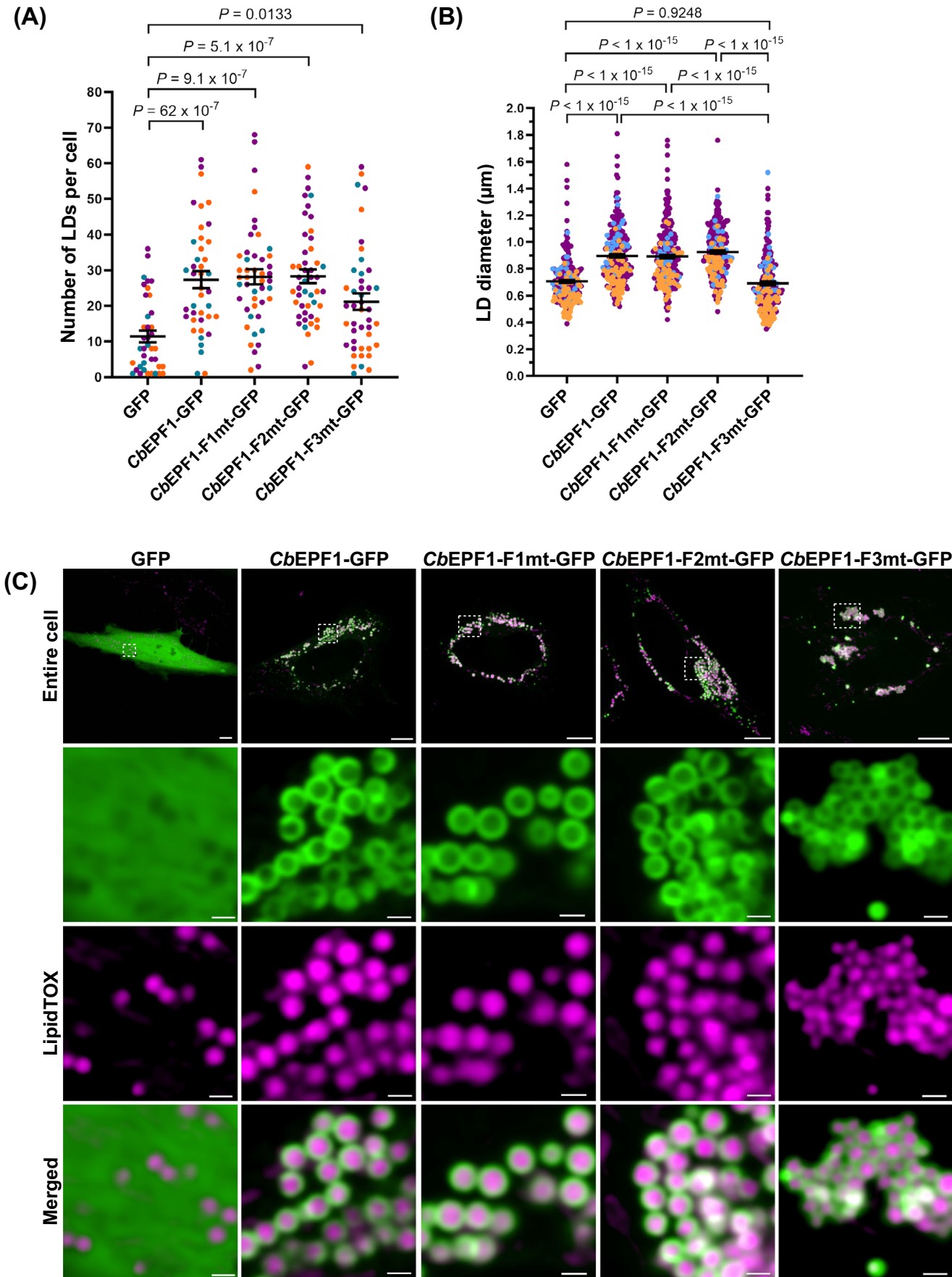

◄ **Figure 7.** *Cb*EPF1 regulates host LD metabolism.

(A) HeLa cells expressing *Cb*EPF1-GFP or *Cb*EPF1-FFAT mutant-GFP contain significantly high number of LDs compared to GFP expressing cells. The number of LDs/cell in HeLa cells transiently expressing respective protein were quantified and shown as mean ± SEM. Data were collected from GFP ($n = 39$), *Cb*EPF1-GFP ($n = 41$), *Cb*EPF1-F1mt-GFP ($n = 46$), *Cb*EPF1-F2mt-GFP ($n = 48$), *Cb*EPF1-F3mt-GFP ($n = 44$) transfected HeLa cells. Cells were randomly selected from three independent experiments. Independent experiments are color-coded. Statistical analysis was performed using one-way ANOVA with Tukey's multiple comparisons test. $P = 62 \times 10^{-7}$ (GFP vs *Cb*EPF1-GFP), $P = 9.1 \times 10^{-7}$ (GFP vs *Cb*EPF1-F1mt-GFP), $P = 5.1 \times 10^{-7}$ (GFP vs *Cb*EPF1-F2mt-GFP), and $P = 0.0133$ (GFP vs *Cb*EPF1-F3mt-GFP). (B) HeLa cells with *Cb*EPF1-GFP or *Cb*EPF1-F1mt-GFP or *Cb*EPF1-F2mt-GFP expression show larger LDs when induced with OA (100 µM) compared to GFP or *Cb*EPF1-F3mt-GFP expressing cells. LD size in cells expressing the respective protein were quantified and shown as mean ± SEM. Data were collected from GFP ($n = 230$ LDs from 13 cells); *Cb*EPF1-GFP ($n = 270$ LDs from 17 cells); *Cb*EPF1-F1mt-GFP ($n = 260$ LDs from 15 cells); *Cb*EPF1-F2mt-GFP ($n = 260$ LDs from 16 cells); *Cb*EPF1-F3mt-GFP ($n = 230$ LDs from 14 cells) transfected HeLa cells. Cells were randomly selected from three independent experiments. Independent experiments are color-coded. Statistical analysis was performed using one-way ANOVA with Tukey's multiple comparisons test. $P < 1 \times 10^{-15}$ (GFP vs *Cb*EPF1-GFP), $P < 1 \times 10^{-15}$ (GFP vs *Cb*EPF1-F1mt-GFP), $P < 1 \times 10^{-15}$ (GFP vs *Cb*EPF1-F2mt-GFP), $P = 0.9248$ (GFP vs *Cb*EPF1-F3mt-GFP), $P < 1 \times 10^{-15}$ (*Cb*EPF1-GFP vs *Cb*EPF1-F3mt-GFP), $P < 1 \times 10^{-15}$ (*Cb*EPF1-F1mt-GFP vs *Cb*EPF1-F3mt-GFP) and $P < 1 \times 10^{-15}$ (*Cb*EPF1-F2mt-GFP vs *Cb*EPF1-F3mt-GFP). (C) Representative images for LD size in HeLa cells expressing GFP or *Cb*EPF1-GFP or *Cb*EPF1-FFAT mutants, treated with OA (100 µM, overnight). Scale bars: 10 µm (overview) and 1 µm (magnified). Source data are available online for this figure.

deciphering the precise mechanisms through which *Cb*EPF1 modulates host lipid metabolism. Additionally, exploring the broader implications of *Cb*EPF1's actions on the overall *C. burnetii* life cycle and pathogenesis could unveil new avenues for understanding the intricacies of host-pathogen interactions.

# Methods

### Reagents and tools table

| Reagent/Resource | Reference or Source | Identifier or Catalog Number |
|---|---|---|
| **Experimental Models** | | |
| HeLa cells | ATCC | CCL-2 |
| Mouse alveolar macrophages (MH-S) | ATCC | CRL-2019 |
| HEK293T cells | ATCC | CRL-3216 |
| **Recombinant DNA** | | |
| pT-Rex-C-GFP plasmid | Invitrogen, CA, USA | 12301016 |
| mCherry-VAPB | Addgene | 108126 |
| pEGFPC1-hVAP-B KD/MD | Addgene plasmid | 104450 |
| pcDNA6.2-M-mCherry-DEST | Invitrogen, CA, USA | 12489027 |
| pEGFPC1-hVAP-B KD/MD | Addgene | 104450 |
| pEFIRES-P-mTagBFP-KDEL | Addgene | 87163 |
| pQCXIP GFP-MOSPD2 WT | Addgene | 186467 |
| pT-Rex-*Cb*EPF1-GFP | This Study | Methods |
| pT-Rex-*Cb*EPF1-F1mt-GFP (Y232A) | This Study | Methods |
| pT-Rex-*Cb*EPF1-F2mt-GFP (F244A) | This Study | Methods |
| pT-Rex-*Cb*EPF1-F3mt-GFP (YF/AA) | This Study | Methods |
| pcDNA6.2-M-mCherry-*Cb*EPF1 | This Study | Methods |
| pKNT-*Cb*EPF1-Wt | This Study | Methods |
| pKNT-*Cb*EPF1-F1mt (Y232A), | This Study | Methods |

| Reagent/Resource | Reference or Source | Identifier or Catalog Number |
|---|---|---|
| pKNT-*Cb*EPF1-F2mt (F244A) | This Study | Methods |
| pKNT-*Cb*EPF1-F3mt (YF/AA) | This Study | Methods |
| pUT18C- VAPB-Wt | This Study | Methods |
| pUT18C VAPB-MSPmt | This Study | Methods |
| **Antibodies** | | |
| Rabbit anti-VAPA | Proteintech | 15275-1-AP |
| Rabbit anti-VAPB | Proteintech | 14477-1-AP |
| Rabbit anti-MOSPD2 | Sigma-Aldrich | ZRB1046 |
| AlexaFluor 568 | Invitrogen | D22912 |
| Goat anti-rabbit IgG-HRP conjugate | Thermo Scientific | PI31460 |
| Anti-mouse IgE-HRP | Invitrogen, CA, USA | SA5-10263 |
| **Oligonucleotides and other sequence-based reagents** | | |
| PCR primers | This study | Methods |
| **Chemicals, Enzymes and other reagents** | | |
| Lipi-Blue | Dojindo Molecular Technologies, Rockville, MD, USA | LD01-10 |
| Roswell Park Memorial Institute (RPMI) 1640 medium | Corning, New York, NY, USA | MT15040CV |
| 10% fetal bovine serum | Atlanta Biologicals, Flowery Branch, GA, USA | S11150 |
| 2 mM l-alanyl-l-glutamine | Glutagro, Corning | 25-015-CI |
| Dulbecco's modified Eagle medium (DMEM) | Corning | MT10101CV |
| FuGENE6 | Promega, Madison, WI, USA | PRE2691 |
| Oleic acid | Sigma-Aldrich, St Louis, MO, USA | O3008-5ML |
| In-Fusion HD cloning | Takara Bio, San Jose, CA, USA | 639648 |
| ProLong Gold | Invitrogen | P36935 |
| HCS LipidTOX Red neutral lipid stain | Invitrogen | H34476 |

| Reagent/Resource | Reference or Source | Identifier or Catalog Number |
|---|---|---|
| GFP-Trap magnetic beads | Bulldog Bio, Inc., Portsmouth, NH, USA | gtma-20 |
| Protease inhibitor cocktail | CST, Danvers, MA, USA | P834 |
| **Software** | | |
| Nikon NIS-elements software | Nikon | |
| Fiji | ImageJ | |
| **Other** | | |
| Nikon Eclipse Ti2 spinning disc microscope | Nikon | |
| Azure 600 western blot imager | Azure biosystems, Dublin, CA, USA). | |

## Cell lines and Bacterial Strains

HeLa cells (ATCC CCL-2; ATCC, Manassas, VA, USA) and mouse alveolar macrophages (MH-S; ATCC CRL-2019, Manassas, VA, USA) were cultured in Roswell Park Memorial Institute (RPMI) 1640 medium (Corning, New York, NY, USA) supplemented with 10% fetal bovine serum (FBS; Atlanta Biologicals, Flowery Branch, GA, USA) and 2 mM l-alanyl-l-glutamine (Glutagro, Corning) at 37 °C in a 5% $CO_2$ atmosphere. HEK293T cells (ATCC CRL-3216) were cultured in Dulbecco's modified Eagle medium (DMEM) with high glucose (Corning) with 10% FBS under similar conditions. All mammalian cell cultures were passaged every 3 days, and no cells older than 20 passages were used in experiments. Regular testing confirmed the absence of mycoplasma contamination in all cell lines. mCherry-expressing *C. burnetii* NMII (Beare et al, 2009) were grown for 4 days in ACCM-2, washed twice with phosphate buffered saline (PBS) and stored as previously described (Omsland et al, 2009). The *E. coli* Δ*cya* strain DHM1, generously provided by Dr. Amanda J. Brinkworth (University of Nebraska Medical Center), was propagated in LB broth (BD Difco, Franklin Lakes, NJ, USA) and plated on LB agar supplemented with appropriate antibiotics.

Cells were transfected using FuGENE6 (Promega, Madison, WI, USA) according to the manufacturer's protocol. Where noted, cells were incubated with 30 or 100 μM oleic acid (Sigma-Aldrich, St Louis, MO, USA) complexed with fatty acid-free BSA for the indicated time to induce LD formation.

## Cloning and constructs

The *Cb*EPF1-GFP expression vector was constructed through PCR amplification using *C. burnetii* genomic DNA as the template with the following primers: forward primer 5′-atccctaccggtgatatgcctga-taaaaccgacagtcttactatac-3′ and reverse primer 5′-ctttgctagccatccgcg-gaactttccaatgtaacgaatccaagtcc-3′. The resulting PCR fragments were cloned into XhoI-linearized pT-Rex-C-GFP plasmid (Invitrogen, CA, USA) using In-Fusion HD cloning (Takara Bio, San Jose, CA, USA).

Mutants of *Cb*EPF1-GFP, including *Cb*EPF1-F1mt-GFP (Y232A), *Cb*EPF1-F2mt-GFP (F244A), and *Cb*EPF1-F3mt-GFP (YF/AA), were generated through site-directed mutagenesis using *Cb*EPF1-GFP as the template (Azenta Life Sciences, South Plainfield, NJ, USA).

The *Cb*EPF1-Wt expression vector was created using PCR amplification with *C. burnetii* genomic DNA as the template and the following primers: forward primer 5′-tgattacgccaagcttgatgcct-gataaaaccgacagtcttactatac-3′ and reverse primer 5′-gcaggcatgcaagc-taacttttccaatgtaacgaatccaagtcc-3′. The PCR fragments were ligated into the HindIII-linearized pKNT plasmid. Similarly, *Cb*EPF1-F1mt (Y232A), *Cb*EPF1-F2mt (F244A), and *Cb*EPF1-F3mt (YF/AA) variants were generated through site-directed mutagenesis (Azenta Life Sciences) using *Cb*EPF1-Wt as the template.

The VAPB-Wt expression vector was produced through PCR amplification using mCherry-VAPB (Addgene plasmid – 108126) as the template and the following primers: forward primer 5′-tac cgagctcgaatttgatggcgaaggtggagcag-3′ and reverse primer 5′-ttatatcgatga attctacaaggcaatcttcccaataattacacca-3′. The PCR fragments were ligated into the EcoRI-linearized pUT18C plasmid. The VAPB-MSPmt expression vector was generated using pEGFPC1-hVAP-B KD/MD (Addgene plasmid – 104450) as the template, harboring mutations in the MSP domain (K87D and M89D), with the same primers and cloning strategy as for VAPB-Wt.

The mCherry-*Cb*EPF1 expression vector was constructed through PCR amplification using *C. burnetii* genomic DNA as the template with the following primers: forward primer 5′-atg gatgagctgtacatgcctgataaaaccgacagtcttact-3′ and reverse primer 5′-tc aaccactttgtacctaaactttccaatgtaacgaatccaagtcc-3′. The resulting PCR fragments were cloned into BsrGI-linearized pcDNA6.2-M-mCherry-DEST plasmid (Invitrogen, CA, USA) using In-Fusion HD cloning (Takara Bio, San Jose, CA, USA).

mCherry-VAPB was a gift from Gerry Hammond (Addgene plasmid # 108126), pEGFPC1-hVAP-B KD/MD was a gift from Catherine Tomasetto (Addgene plasmid # 104450), pEFIRES-P-mTagBFP-KDEL was a gift from Elina Ikonen (Addgene plasmid # 87163), pQCXIP GFP-MOSPD2 WT was a gift from Catherine Tomasetto (Addgene plasmid # 186467). All plasmid constructs were validated through Sanger dideoxy sequencing (ACGT, Inc., Wheeling, IL, USA) and Oxford Nanopore Technology (Plasmid-saurus, Eugene, OR, USA).

### Microscopy

All experiments were conducted by live cell imaging on a Nikon Eclipse Ti2 spinning disc microscope within an environmental chamber (5% $CO_2$ and 37 °C). Cells transfected with plasmids were cultured on ibidi slides (ibidi USA, Inc., Fitchburg, WI, USA), and imaged 48 h post-transfection. For experiments involving oleic acid treatment, growth media was supplemented with 100 μM oleic acid for 16 h before imaging. To observe the association of *Cb*EPF1 with LD biogenesis, time-lapse movies were acquired at 1-min intervals for 20 min. Data quantification and image processing were carried out using the Nikon NIS-elements software. The Alpha-blending algorithm in NIS elements software was used for 3D rendering.

## Immunofluorescence

Cells were grown on glass coverslips, fixed in 2.5% paraformalde-hyde (PFA) in PBS for 15 min, and permeabilized with 0.1% saponin in PBS for 15 min. After blocking with 1% bovine serum albumin in PBS (PBS-BSA), cells were incubated for one hour at room temperature with the primary antibody in PBS-BSA. Primary antibodies were: rabbit anti-VAPA (1:250; Proteintech, 15275-1-AP) and rabbit anti-VAPB (1:250; Proteintech, 14477-1-AP). Cells

were washed three times in PBS and incubated for 30 min with the secondary antibody (AlexaFluor 568; Invitrogen, D22912). After three washes with PBS, the cells were incubated with Lipi Blue (1:1000; Dojindo Molecular Technologies, LD01) for 30 min at room temperature. Cells were washed with PBS once, coverslips mounted in ProLong Gold (Invitrogen), and imaged using a spinning-disk confocal microscope (Nikon Eclipse Ti2).

## Transmission electron microscopy

MH-S cells (mouse alveolar macrophages) were infected with mCherry-*C. burnetii*. At two days post-infection, infected cells were sorted using the mCherry signal. Uninfected or *C. burnetii*-infected macrophage cells were fixed (2% glutaraldehyde, 2% paraformaldehyde, 0.1 M phosphate buffer) overnight at room temperature. The fixed cells were collected by centrifugation ($1000 \times g$) at room temperature. After three washes with 0.1 M phosphate buffer, the cells were post-fixed in 1% Osmium tetroxide ($OsO_4$) for 30 min and then washed three times with 0.1 M phosphate buffer. Following washing, the cells were further dehydrated by a graded series of ethanol solutions (50%, 70%, 90%, 95%, and 100%) and processed for embedding in epoxy resin (Embed812). Ultrathin (70-nm) sections were collected on 200-mesh copper grids, stained with 1% uranyl acetate and lead citrate, and observed by transmission electron microscopy. A 120 kV electron microscope (FEI Tecnai G2 Spirit TEM) was used at 80 kV. Images were captured with an AMT CCD camera using Digital Micrograph software at room temperature. ER-LD contacts were evaluated as previously described (Lam et al, 2021). Briefly, the percentage of lipid droplet surface covered with endoplasmic reticulum was measured by calculating the perimeter of lipid droplet and length of endoplasmic reticulum that is close (<40 nm) to lipid droplet surface using ImageJ software (Schindelin et al, 2012).

## Bacterial adenylate cyclase-based two-hybrid assay (BACTH)

The BACTH assay was performed as previously described (Ouellette et al, 2017). Briefly, the *E. coli* Δ*cya* strain DHM1 was co-transformed with pKNT25 and pUT18C derivatives. Following transformation, the bacteria were plated on LB agar supplemented with 5-bromo-4-chloro-3-indolyl-β-d-galactopyranoside (X-gal, 40 µg ml$^{-1}$), IPTG (1 mM), Carbenicillin (100 µg ml$^{-1}$), and Kanamycin (50 µg ml$^{-1}$) and incubated for 48 h at 30 °C. Blue color of the colonies indicate a positive interaction between the analyzed proteins, while white color denotes the absence of complementation.

## Immunoprecipitation

HEK293T cells were seeded at a density of $1 \times 10^5$ cells per well in a 6-well plate and transfected with GFP or *Cb*EPF1-GFP or *Cb*EPF1-F1mt-GFP or *Cb*EPF1-F2mt-GFP or *Cb*EPF1-F3mt-GFP, using Fugene6 transfection reagent (Promega). After 24 h of transfection, cells were treated with 100 µM oleic acid for 16 h. Subsequently, cells were collected and lysed on ice for 30 min using an immunoprecipitation (IP) buffer (15 mM Tris pH 7.4, 150 mM NaCl, 1% Triton X-100, 1x protease inhibitor cocktail (CST, Danvers, MA, USA). The supernatant obtained after centrifugation at $13,000 \times g$, 4 °C for 10 min was incubated with GFP-Trap magnetic beads (Bulldog Bio, Inc., Portsmouth, NH, USA) at 4 °C overnight. Following the incubation, the beads were washed with

the IP buffer and bound proteins eluted and analyzed using SDS–polyacrylamide gel electrophoresis (SDS-PAGE) followed by immunoblotting with appropriate antibodies.

## Western blot analysis

Proteins were separated on 4–20% Mini-PROTEAN TGX stain-free protein gels (Bio-Rad, Hercules, CA, USA) and transferred to a nitrocellulose membrane. The membrane was blocked with 5% skimmed milk in TBS-T for 1 h at room temperature. The membranes were washed with 1x TBST, incubated with primary antibodies in 1% BSA-TBST overnight at 4 °C, and then washed with 1x TBST prior to incubation with secondary antibodies coupled to horseradish peroxidase. Signals were detected using an enhanced chemiluminescence system (Thermo Fisher Scientific, Inc., Waltham, MA, USA) and visualized with Azure 600 western blot imager (Azure biosystems, Dublin, CA, USA). The primary antibodies used were anti-GFP (1:1000; Sigma-Aldrich, G6539), anti-MOSPD2 (1:1000; Sigma-Aldrich, ZRB1046), anti-VAPA (1:2000; Proteintech, Rosemont, IL, USA, 15275-1-AP), and anti-VAPB (1:1000; Proteintech, Rosemont, IL, USA, 14477-1-AP). Secondary antibodies used were goat anti-mouse IgE-HRP conjugate (1:10,000, Invitrogen, SA5-10263) and Goat anti-rabbit IgG-HRP conjugate (1:10,000; Thermo Scientific, PI31460).

## LD measurements

Lipi-Blue (Dojindo Molecular Technologies, Rockville, MD, USA) and HCS LipidTOX Red neutral lipid stain (Invitrogen) were used for LD staining. LD measurements were performed as previously described (Li et al, 2019). LD numbers/cell were manually quantified in the absence of oleic acid treatment using Nikon NIS Elements AR software. LD diameter was measured after treatment with 100 µM oleic acid using Nikon NIS Elements AR software. The cells or images for LD analysis were chosen randomly from three independent experiments.

## Statistical analysis

The statistical parameters, including n, mean and SEM, are reported in the corresponding figure legends. Statistical analysis was performed in Prism (GraphPad, San Diego, CA, USA). Statistical significance was calculated by one-way ANOVA analysis or two-tailed unpaired Student's *t*-test. *P* values were reported in figures and figure legends (Alpy et al, 2013; Olarte et al, 2022; Salo et al, 2016; Schindelin et al, 2012).

# Data availability

Source files for microscopy data are available at BioImage Archive ID: S-BIAD1302 (https://doi.org/10.6019/S-BIAD1302) and for expanded view figures at BioImage Archive ID:S-BIAD1300 (https://doi.org/10.6019/S-BIAD1300).

The source data of this paper are collected in the following database record: biostudies:S-SCDT-10_1038-S44319-024-00266-8.

## Peer review information

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

## Acknowledgements

We thank Dr. Micah Schott for critical comments on the manuscript and members of the Gilk Lab for helpful discussions and critical feedback. We thank Dr. Amanda Brinkworth for providing plasmids for BACTH assay and Nicholas Conoan and Deepti Negi of the Electron Microscopy Core Facility (EMCF; RRID:SCR_017864) at the University of Nebraska Medical Center for technical assistance. The EMCF is supported by state funds from the Nebraska Research Initiative (NRI) and the University of Nebraska Foundation, and institutionally by the Office of the Vice Chancellor for Research. This work was supported National Institutes of Health (AI139176 to SDG) and the American Heart Association (906475 to RKA). Synopsis image was created using pictures from Servier Medical Art, by Servier (http://smart.servier.com) and Fig. 3A was created using Biorender software (http://Biorender.com).

## Author contributions

**Rajendra Kumar Angara**: Conceptualization; Data curation; Formal analysis; Funding acquisition; Investigation; Methodology; Writing—original draft; Writing—review and editing. **Arif Sadi**: Formal analysis; Investigation; Methodology; Writing—review and editing. **Stacey D Gilk**: Conceptualization; Formal analysis; Supervision; Funding acquisition; Methodology; Writing—original draft; Project administration; Writing—review and editing.

Source data underlying figure panels in this paper may have individual authorship assigned. Where available, figure panel/source data authorship is listed in the following database record: biostudies:S-SCDT-10_1038-S44319-024-00266-8.

## Disclosure and competing interests statement

The authors declare no competing interests.

# Expanded View Figures

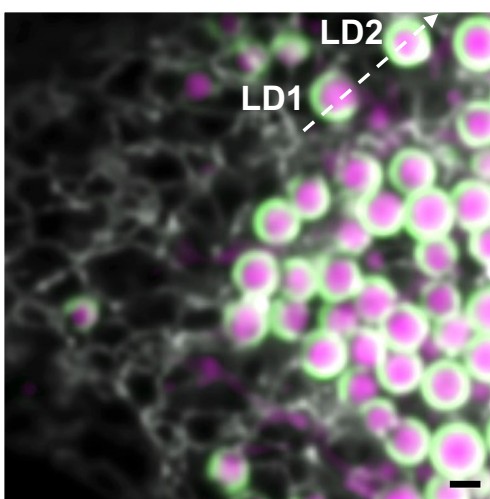
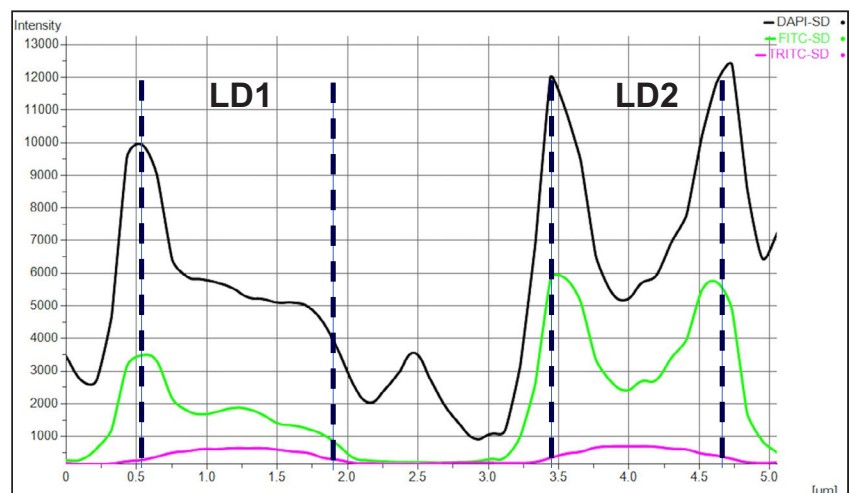

**Figure EV1.  *Cb*EPF1-GFP and BFP-KDEL localization along lipid droplets.**

Intensity profiles of *Cb*EPF1-GFP and BFP-KDEL along the two lipid droplets (dotted lines). The shown image is from Fig. 2B bottom panel, which was reused for representation and intensity profile analysis. Scale bar on merged image: 1 µm. Intensity profiles: Black—BFP-KDEL, Green—*Cb*EPF1-GFP, and Magenta—LipidTOX red.

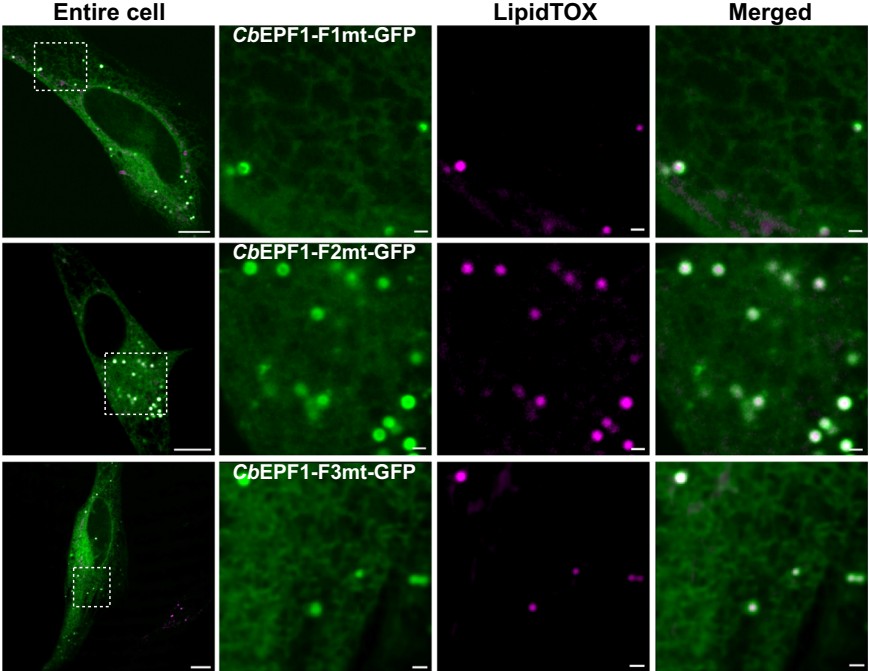

**Figure EV2. Localization of *Cb*EPF1-FFAT mutant-GFP in HeLa cells.**

The ectopically expressed *Cb*EPF1-F1mt-GFP or *Cb*EPF1-F2mt-GFP or *Cb*EPF1-F3mt-GFP localized to ER and LDs in HeLa cells. The *Cb*EPF1-FFAT mutant-GFP proteins showed a similar localization pattern to *Cb*EPF1-GFP (Fig. 1B). The LDs were labeled with LipidTOX-Red. Scale bars: 10 μm (overview) and 1 μm (magnified).

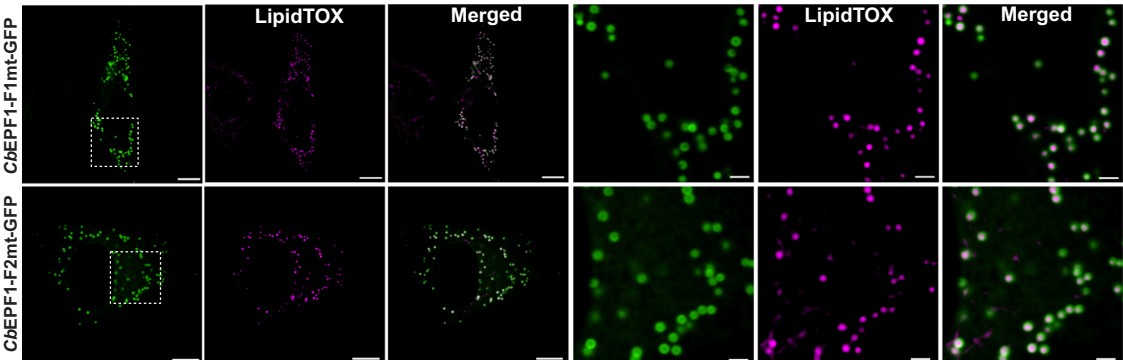

**Figure EV3.  Dispersion of LDs in cells expressing *Cb*EPF1-F1/F2 mutant-GFP.**

Cells expressing *Cb*EPF1-F1mt-GFP or *Cb*EPF1-F2mt-GFP exhibit dispersed LDs in the HeLa cells. Scale bars: 10 µm (overview) and 1 µm (magnified).

