## [Peer Review File · EMBO Reports]

A novel bacterial effector protein mediates ER-LD membrane contacts to regulate host lipid droplets

Rajendra Angara, Arif Sadi, and Stacey D. Gilk

Corresponding author(s): Stacey D. Gilk (sgilk@unmc.edu)

Review Timeline:

Transfer Date:	27th Mar 24
Editorial Decision:	4th Apr 24
Revision Received:	2nd Aug 24
Editorial Decision:	26th Aug 24
Revision Received:	5th Sep 24
Accepted:	10th Sep 24

Review
COMMONS

Transaction Report: This manuscript was transferred to EMBO reports following peer review at Review Commons.

Review #1

1. Evidence, reproducibility and clarity:

Evidence, reproducibility and clarity (Required)

****Summary:****

The authors report a *Coxiella burnetii* effector protein, CbEPF1, which associate with lipid droplets (LD) at points of contact with the endoplasmic reticulum (ER). The presence of two FFAT motifs in CbEPF1 mediates CbEPF1 interaction with the ER protein VAP, LD-ER interaction, and an increase in LD size. Based on these results the authors put forward a model by which translocation of CbEPF1 into the host cell cytosol results in regulation of host cell lipid metabolism via the formation of ER-LD contact.

The study relies heavily on fluorescence microscopy of ectopic (over)expression of CbEPF1 in eukaryotic cells.

****Major comments:****

CbEPF1 and ER-LD contact:

The immunofluorescence analysis of cell ectopically expressing CbEPF1-GFP provides convincing evidence that CbEPF1-GFP associates with LD and that CbEPF1-positive LD are positive for ER markers such as BFP-KDEL and VAPB (Fig 1&2). However, the study would benefit from looking at endogenous proteins to rule out any potential overexpression artefacts. Does endogenous VAP localize to CbEPF1-positive LD? Does CbEPF1 expressed from *Coxiella* (endogenous protein, or at least a tagged protein expressed from the bacteria) localize to LD?

While the immunofluorescence images are of very high quality and convincing, electron microscopy is necessary to ascertain that membrane contacts between LD and the ER are induced upon CbEPF1 overexpression.

CbEPF1 interaction with the ER is linked to the FFAT motifs (see below); what is known about CbEPF1 association with LD? Can a specific CbEPF1 domain be identified? In other words, does CbEPF1 contains 2 distinct membrane targeting domains that confer specificity to each of the contacting organelle (ER and LD in the present case) and thereby resemble other contact site localizing proteins.

CbEPF1 FFAT motifs and VAP binding:

The similarity of the sequence of the CbEPF1 FFAT motifs to the canonical sequence (Fig 1A), combined with data with alanine substitution mutation of the essential residue in position 2 of the FFAT motifs (Fig 3, 4), strongly support that CbEPF1 contains 2 functional FFAT motifs that confer VAP binding.

FFAT motifs can mediate binding to VAPA, VAPB, and/or MOSPD2. Moreover, in addition to forming homodimer, VAPA, VAPB, and MOSPD2 can form heterodimers. What is the rationale for using VAPB

(Fig 3CDE)? Do VAPA and/or MOSPD2 also yield positive results using the assays performed in Fig 3CDE, or is the CbEPF1-VAP interaction specific to VAPB? In the same line, in Fig 3E, is it possible that the CbEPF1-MOSPD2 is indirect and due to VAP- MOSPD2 interaction?

Regarding the two FFAT motifs: are the two FFAT motifs redundant or synergistic? Although the data is not quantified, Figure 3E suggest a synergistic effect for VAP binding, however most IF data suggest redundancy. On the other end, the second FFAT motif seems necessary for MOSPD2 binding.

Overall, clarifying CbEPF1 binding specificity towards a VAP/MOSPD2 and the role of each FFAT motif in this process could elevate the study by providing mechanistic insights into the hijacking of ER-LD contact sites by Coxiella and comparing and contrasting with the formation of ER-LD in naïve cells and/or the hijacking of VAP-dependent contact by other microbial pathogens.

CbEPF1-mediated LD clustering in the absence of VAP binding

A CbEPF1 FFAT motif mutant (F3mt) associates with LD that do not associate with the ER marker KDEL, and causes LD clustering (Fig 4). The authors speculate that the lack of LD-ER interaction results in LD-LD interaction, potentially via interaction of unidentified protein(s) on the LD surface. What is the biological relevance of the LD clustering phenotype? What is known about the role of LD clustering vs ER-LD contact, in the context of lipid metabolism? Could mechanistic characterization of this phenomenon provide insights in LD biology and/or the role of CbEPF1/ER-LD contacts in the context of Coxiella infection?

CbEPF1-mediated increase in LD number and size

CbEPF1-GFP overexpression result in an increase in the number of LD per cell independently of the FFAT motifs (Fig 5A). CbEPF1-GFP overexpression also result in an increase in LD diameter; however, this phenotype is dependent on wild-type FFAT motifs (Fig 5B).

Quantification and corresponding statistical analysis support these conclusions. However, the representative images are not necessary in line with the bar graphs. For example, there appear to be less LD upon expression of F1mt or F2mt, compared to WT. Additionally, the increase in size is moderate and hard to appreciate in the IF images. It is also unclear, if/how an increase in LD number and/or size is biologically relevant in the context of Coxiella infection.

Regarding potential mechanism(s), it is also unclear how CbEPF1 is promoting an increase in LD number. De novo LD production at the ER is unlikely given that the FFAT motifs, and therefore ER association, are not required. What would an alternative mechanism be, and can it be experimentally tested?

Regarding the increase in LD size, the author suggest that the phenotype could be due to impaired lipid transfer from the ER to LD. This is an interesting model. Do the authors envision that CbEPF1 is a lipid transfer protein and/or act on ER-LD associated lipid transfer? Can either be experimentally tested?

Altogether, strengthening this aspect of the study would clarify the proposed model and significantly increase the impact of the study.

Functional relevance:

One aspect that is not addressed in the study, is what are the benefit(s), if any, of CbEPF1 translocation into the host cytosol and targeting to ER-LD contact? This could be addressed by assessing the phenotype of a Coxiella CbEPF1 mutant? On the flip side, is VAP required for Coxiella intracellular growth/vacuole maturation? Another avenue would be to investigate if CbEPF1 affects the lipid composition of the CCV. The present study suggest that LD may be important for Coxiella intracellular life cycle. Are LD formation induced upon infection? Are ER-LD contact increased upon infection? One may also expect that inhibition of LD formation would affect bacterial replication and that stimulation may promote growth/vacuole maturation? Any experiments addressing the biological relevance of the present findings in the context of Coxiella infection would tremendously increase the impact of the study.

****Minor comments:****

None. The manuscript is well written and easy to follow.

2. Significance:

Significance (Required)

Inter-organelle communication through the formation of membrane contact between two apposing organelles is critical to maintain host cell homeostasis via the transfer of small molecules such as lipid and ions. Bacterial and viral pathogens have been shown to manipulate proteins that localize to cellular membrane contact and to promote membrane contact between their intracellular vacuole and the ER. Both viral and bacterial proteins that contains FFAT motifs and interact with VAP have been described in the context of tethering of the ER to the membrane compartment in which the pathogen replicate.

The present study stands out by the characterization of a FFAT motifs-containing bacterial effector protein that targets cellular contact, ER-LD more specifically. The significance is 2-fold. It expands the list of FFAT-motif containing proteins in pathogens, reinforcing the idea that VAP/membrane contact may be a universal mechanism of host-pathogen interaction, which will be of interest to those studying host-microbe interaction in basic or translational research settings. The study also has the potential to move forward research on LD biology and LD-ER contact, which will be of interest to cell biologists in general, and to the evergrowing membrane contact community, specifically.

3. How much time do you estimate the authors will need to complete the suggested revisions:

Estimated time to Complete Revisions (Required)

(Decision Recommendation)

Between 3 and 6 months

4. Review Commons values the work of reviewers and encourages them to get credit for their work. Select 'Yes' below to register your reviewing activity at Web of Science Reviewer

Recognition Service (formerly Publons); note that the content of your review will not be visible on Web of Science.

No

Review #2

1. Evidence, reproducibility and clarity:

Evidence, reproducibility and clarity (Required)

This is a strong manuscript about the existence of proteins coded by intracellular parasites (here Coxiella) that have evolved to parasitise the lipid transport machinery of their hosts. This is a first in that the parasite protein acts at a distance from the parasite itself, manipulating two of the host organelles - and not acting at their site of contact with PVs. There is considerable research into one protein and its effect when expressed by itself.

Despite all the advances there are a couple of areas where the manuscript can be improved, and a few extra fairly straightforward experiments added about the amphipathic helix. Even though these are unlikely change the overall message, they would make the story more complete.

****Major points****

More details are required about the amphipathic helix. Check that the AH does target LDs by expression of the AH alone in a GFP chimera +/- oleate and then mutagenesis. Also show the AH in a helical wheel projection (eg by Heliquest) and say if it aligns with similar AHs in homologs (see my point below)

Fig 1B: In infected cells, do the affected LDs tend to be close to the PVs?

Also in Fig 1B: highlight small KDEL+ve ER rings around LDs here. Study whether LDs have these in infected cells without the confounder (?artefact) of EPF1 over-expression

Fig 2A the ER looks quite different here from Fig 1B, even at t0. Grossly the strands are spaced wider apart. In detail there are no rings around LDs. Can the authors explain this? Which morphology is common, especially in cells early in infection without co-expressed protein?

Fig 6 & Line 237: "As the N-terminal region of CbEPF1 is undefined": I suggest that the authors could do more here. At minimum change model to highlight the strong probability that the N term is a globular domain that functions at the LD ER interface. (What are three other unidentified LD proteins? I suggest omitting them).

Although the Alphafold prediction for EPF1 is low confidence only, in a few minutes of BLAST searching I found the homolog A0A1J8NR10_9COXI (also FFAT+ve) which has a moderately confident structural prediction for its N terminus. This model has a quite large internal hydrophobic cavity, indicating lipid transfer capability and function similar to known LTPs. This means that action as a "tether" possibly results from experiments with viral promoters (see minor point on terminology).

****Minor:****

Fig 2B: add more arrowheads/arrows to fit legend (says they are both multiple)

FFAT selectivity for MOSPD2: say if this fits the di Mattia or (as appears likely) it extends the known differences between VAPA/B and MOSPD2. Also say if VAPA is expected to behave as VAPB

Explain how "Mutations in the CbEPF1 FFAT motif(s) did not influence CbEPF1-GFP localization to either the host ER (Supplementary Fig. 1)". In F3mt this shows that EPF1 has a way to target ER other than FFAT/VAP. Discuss if that is via AH insertion in ER.

Also, the (admittedly low) level of ER targeting is possibly slightly reduced by F3mt, as shown by greater GFP in the nucleus in the single cells shown. If this is a feature of the whole field of cells, it implies that the FFATs normally work with the AHs to target EPF1 to the ER.

"clustering" LDs w F3mt: could this indicate dimer formation by CbEPF1? Note: to me it appears wrong to describe fig 4A as showing ER exclusion. LD proximities to each other dominate. It's not 100% clear that LDs cluster as their proximities are not universal: "LD-LD interactions" may be (very) weak.

Fig 5: can levels of EPF1 here be compared to those in cells undergoing natural infection (approximate comparison by qPCR better than nothing if no antibodies are available)? Fig 5a: would it be possible to increase the number of cells counted to attempt to make the reduced number of LD in F3mt significant?

****Minor****

Line 226: no sequence homology: misses the point- there is the common feature of an AH

Issue to be discussed, as probably too difficult to experiment on: when EPF1 is on the ER does it engage vap only weakly (implying a means to mask its motifs), since if the interaction is strong vap is then unable to bind other partners?

Line 245: "MOSPD2, a sole VAP that is known to localize on LD surfaces" (worth citing Zouiouich again here). Do the cells/tissues infected by Coxiella express MOSPD2?

Line 259/260: this "suggestion" about cholesterol should be toned down. It is a speculation that could be tested in future, but the data here do not suggest it.

'Tether' this word implies more than just bridging but also a role in the physical formation of the contact. Since EPF1 most likely has an LTP domain, it seems linguistically confusing to refer to it as a tether, especially since the experiments that physically later LD-ER contact involve probable over-expression.

Discuss whether it is 100% certain that EPF1 is in the host cytosol or whether some experiment(s) at a future date (proteomic/western blotting) will be needed to make that conclusion 100% secure.

****Referees cross-commenting****

COMMENT 1

I realise that both reviewer 1 and reviewer 3 have considered this MS carefully, but I think that their reviews could be improved in some respects. I will add two comments, one for each of the other reviews.

Reviewer 1.

The review poses multiple questions to the authors suggesting that answering these questions experimentally would strengthen the paper. Some of the points seem to misunderstand what is the accepted standard for membrane cell biological research into membrane contact sites. While it might be that the authors can rebut these points, I think it is preferable to use Cross Commenting as an opportunity to address these issues beforehand.

Major Comment 1: CbEPF1 and ER-LD contact

Looking at endogenous proteins:

I wondered about the same point, but I concluded that this is not likely to be possible in the scope of this submission. If it were possible then I guess the authors would have attempted it. Looking on Google Scholar I could find no example of an endogenous Coxiella proteins being tagged in the bacterial genome. So the only way to find the portion is via an antibody. Assuming the authors do not have one, I do not think we should ask for one at this stage in the publication process.

Electron microscopy:

the reviewer is incorrect to say that this is necessary. It may be the gold standard, but it is a huge amount of extra work. Furthermore it is not at all necessary when the protein in question localises clearly to the interface between organelles identified by confocal microscopy.

Can a specific CbEPF1 domain be identified?

Here a Amphipathic Helix has been identified, but the lack of dissection of that region by the

authors explains this question by the reviewer, which is also shared by Reviewer 3. I agree with the implication that more should be done to dissect that.

Major Comment 2: CbEPF1 FFAT motifs and VAP binding

Are the two FFAT motifs redundant or synergistic?

I would say that the authors have addressed that to a reasonable extent

CbEPF1 binding specificity towards a VAP/MOSPD2

Ditto

Major Comment 3: LD clustering

Since this is an effect of mutated protein only, I think that the 3 questions posed at the end here need only be addressed in Discussion.

Major Comment 4: CbEPF1-mediated increase in LD number and size

less LD upon expression of F1mt or F2mt, compared to WT:

this seems wrong. The numbers are the same. The comment about IF images are unjustified as they have been quantified and do show a difference.

I agree that the biological relevance is unclear, and that this might be addressed. That would require making a mutant Coxiella strain. While that would make a big difference to this work, my feeling is that this is well over a year's work. I would be guided by the authors on that and I would not suggest it as required for this MS.

De novo LD production at the ER is unlikely:

This statement is ill-considered as the FFAT motifs ARE required (Fig 5). Furthermore, in all systems ever reported de novo LD production takes place at the ER, so any alternative would be quite extraordinary.

Altogether, strengthening this aspect of the study:

In my view, this area does not need more work and it would not be constructive to ask for more.

Major Comment 5: Functional relevance

assessing the phenotype of a Coxiella CbEPF1 mutant

I agree that this would be good, but it mightn't be feasible within the confines of this one paper. In the various projects that have made transposon mutants of Coxiella, has a strain been made that affects EPF1? If not, then the authors should state this and discuss it as work for the future. The

reviewers cannot expect any experiments!

Is VAP required for Coxiella intracellular growth/vacuole maturation?

On the surface this suggestion seems to offer an experimental route to understanding EPF1. However, VAP binds to >100 cellular proteins, many relating to lipids traffic and a considerable number of these already localised to lipids droplets (ORP2, MIGA2, VPS13A/C). It is therefore unlikely that such an experiment would be interpretable, and I recommend that this request be reconsidered.

Are LD formation induced upon infection? Are ER-LD contact increased upon infection?

These are very reasonable ideas and the results would be interesting additions to this paper.

COMMENT 2

I have given one set of comments already. Here are my comments for Reviewer 3.

The review makes a few assumptions that I question. While it might be that the authors can rebut these assumptions, I think it is preferable to use Cross Commenting as an opportunity to address these issues beforehand.

Major Point 1:

What is surprising is that the BFP-KDEL signal is also localizing to the LD surface:

"Surprising" is misguided, as it seems to deny the probability that there is a class of proteins that sit at organelle interfaces binding to both partners simultaneously. Maybe the reviewer means "significant" here, in which case I would agree.

The authors must perform LD isolation

the reviewer is incorrect to say that this must be done. It is a huge amount of extra work. Furthermore it is not at all necessary when the protein in question localises clearly to the structures, and it may not even work as the protein may need a reasonably high general concentration to avoid gradual dissociation (with any re-association) during organelle purification.

what features of the protein enable its LD binding?

Here an Amphipathic Helix has been identified, but the lack of dissection of that region by the authors explains this question by the reviewer, which is also shared by Reviewer 1. I agree with the implication that more should be done to dissect that.

Major Point 2: Quantitative image analysis:

Mander's Colocalization analyses with Costes correction are required

No. The images in Figure 4 speak for themselves.

Please show the LD phenotype of untransfected, and CbEPF1-GFP transfected cells also

This is a good idea.

provide a means to quantify the clustering of LDs
Unnecessary. Not all findings need to be quantified.

Major Point 3:

Data depends largely on overexpression of the protein in uninfected cells.
I agree

What is the localization of the protein in infected cells?

I wondered about the same point, but I concluded that this is not likely to be possible in the scope of this submission. If it were possible then I guess the authors would have attempted it. Looking on Google Scholar I could find no example of an endogenous Coxiella proteins being tagged in the bacterial genome. So the only way to find the portion is via an antibody. Assuming the authors do not have one, I do not think we should ask for one at this stage in the publication process.

What happens to ER-LD contacts upon infection with *C. burnetii*?

This is a very valid question, and answering it would not only strengthen the manuscript but should be achievable in 1-3 months.

2. Significance:

Significance (Required)

This work takes a reasonably big step towards uncovering how parasites have mimicked the molecular machinery of contact sites, here in the context of ER-LD interactions and tantalizingly suggestive of lipid transfer at that contact site (although hard to get strong evidence for that at this stage). This provides yet more evidence for the conservation and overall importance to cells of lipid transfer at contact sites, as well as reminding us of the ability of parasites to attack every aspect of cell function.

3. How much time do you estimate the authors will need to complete the suggested revisions:

Estimated time to Complete Revisions (Required)

(Decision Recommendation)

Between 1 and 3 months

4. Review Commons values the work of reviewers and encourages them to get credit for their work. Select 'Yes' below to register your reviewing activity at Web of Science Reviewer Recognition Service (formerly Publons); note that the content of your review will not be visible on Web of Science.

Yes

Review #3

1. Evidence, reproducibility and clarity:

Evidence, reproducibility and clarity (Required)

Angara et al describe a secreted *Coxiella burnetii* effector with an FFAT motif, which they name CbEPF1, that localizes to host cell LDs and is able to interact with VAP domain containing host proteins. Furthermore, this interaction is able to impact LD size. The study has several interesting findings that suggest that this protein can impact ER structuring around lipid droplets. The use of bacterial 2-hybrid approach to demonstrate binding between CbEPF1 and VAP domain containing proteins is convincing, further supported by co-ip experiments that go on to establish specificity of each of the FFAT motifs for distinct VAP domain containing proteins. However, the relationship between LD localization of the protein and ER-restructuring to the outcome of infection is not clear from the data presented here. In addition, there are many analyses that need to be performed for the data to convincingly support the claims made in the manuscript.

My major concerns are:

1. The authors claim that CbEPF1 localizes to lipid droplets in *Coxiella burnetii* infected epithelial cells. Towards this, the authors express CbEPF1-GFP in *C. burnetii* infected cells expressing BFP-KDEL. The data in Figure 1B and 1C indicate that the protein localizes to lipid droplets in oleic acid treated cells. It is interesting to note that without addition of oleic acid, the protein localizes to the ER. What is surprising is that the BFP-KDEL signal is also localizing to the LD surface (Figure 1B and 2A). While in this section it seems that the protein migrates from ER to LDs, in the later sections, similar data is used to make the claim that the protein induces ER apposition to the LD and localizes to regions of LD-ER contact. Therefore, it raises several questions about the localization of the protein: (i) is it an ER-localized protein that migrates to LDs. In that case, what features of the protein enable its LD binding? In addition, the authors must perform LD isolation to validate that the protein indeed localizes to lipid droplets. (ii) Is it an ER-localized protein, that like BFP-KDEL has the ability to localize to ER-LD contact sites, but remains on the ER membrane? Again, biochemical evidence supporting membrane specification is important to understand the localization of the protein. The authors have focussed only on the FFAT motifs of CbEPF1 and not described the overall domain analysis of the protein. It is therefore difficult to understand at this point how the protein localizes to the ER and the ER-LD junction or LD.

2. Quantitative image analysis:

(i) Authors must perform colocalization analyses to substantiate the claims for ER/LD localization. The authors refer to "extended ER-LD contacts" in figure 2B and the text. These data need to be supported with colocalization analysis between BFP-KDEL and the GFP channel.

(ii) Related to Figure 4A: Mander's Colocalization analyses with Costes correction are required to convincingly demonstrate that the dual FFAT motif is required for ER-LD contacts.

(iii) Related to Figure 4B: Please show the LD phenotype of untransfected, and CbEPF1-GFP

transfected cells also. Can the authors provide a means to quantify the clustering of LDs.

(iv) Figure 5A and B. It is not clear from the figure legend whether the data are pooled from multiple experiments or a single experiment. Experimental replicates must be incorporated in the final analysis.

3. The data presented in this manuscript depends largely on overexpression of the protein in uninfected cells. Given that *C. burnetii* induces LD formation, there are three main areas that need clarity:

(i) What is the localization of the protein in infected cells without the addition of oleic acid under conditions where infection itself induces LD biogenesis.

(ii) What happens to ER-LD contacts upon infection with *C. burnetii*?

(iii) Does the presence of CbEPF1 play any role in infection induced LD biogenesis? This may be an optional experiment to undertake at this point as this would involve significant amount of time investment in generating bacterial strains.

****Minor comments:****

1. Line 36: "maintain" instead of "maintains"

2. The introduction cites mainly reviews with overlapping concepts but does not cite primary literature in the area of organelle-organelles contacts and inter-organelle communication (lines 39-40 and line 49). It would be good to cite key primary research articles in these areas.

3. There are some crucial references related to bacterial secreted effectors that target host lipid droplets, that are missing from the introduction. For example, *Chlamydia trachomatis* is known to secrete effectors that localize to host lipid droplets (PMID: 18591669). *Legionella pneumophila* secretes a small GTPase LegA15 to lipid droplets impacting vesicle secretion of the host cell (PMID:36525490).

4. For all figures, please show all individual channels in monochrome and a merge of BFP-KDEL+LD marker and merge of CbEPF1-GFP+LD marker and/or BFP-KDEL+CbEPF1 (wherever appropriate).

2. Significance:

Significance (Required)

The study by Angara et al reports a dual FFAT motif containing protein of *Coxiella burnetii* that impacts ER-LD association. The strengths of the study lie in characterization of the FFAT motifs in VAP domain binding, and the role of these FFAT motifs in mediating ER-LD contacts. However, the claims made towards LD localization versus localization to ER-LD contact sites by this protein are not well supported by the data. In addition, the relevance of these findings in infected cells are not addressed in this study as the data presented here pertain to an over-expression system.

Bacterial proteins that exit the bacterial containing vacuole and impact organelles outside the endocytic compartments are fascinating as they have the potential to impact global processes such as transcription, metabolism, and protein secretion. Lipid droplet (LD) homeostasis is

dysregulated in many bacterial infections and also plays a crucial role in host defense against infection. Therefore, the knowledge of bacterial effectors in this dysregulation will also potentially provide means of countering bacterial strategies to affect host LD homeostasis. Therefore, the findings presented by Angara et al will provide conceptual and mechanistic advances to the specialized audience in infection and immunity. However, I can foresee that the findings have the potential for interesting tools to be developed for organelle-organelle contacts to be studied, provided there is more clarity on how the protein localizes to the ER/ER-LD contacts/LDs.

3. How much time do you estimate the authors will need to complete the suggested revisions:

Estimated time to Complete Revisions (Required)

(Decision Recommendation)

Between 3 and 6 months

Yes

Revision Plan

Manuscript number: RC-2023-02299

Corresponding author(s): Stacey Gilk

1. General Statements

Intra-organelle communication is critical to maintain cellular homeostasis, where tether proteins play a pivotal role in establishing membrane contact sites between cellular organelles. This is true in the context of host-pathogen interactions, where examples are emerging of pathogens utilizing membrane contact sites to promote infection. Previous studies identified membrane contact sites between bacteria-containing vacuoles (BCVs) and the host endoplasmic reticulum (ER). Indeed, we previously demonstrated the presence of membrane contact sites between the host ER and the BCV of *Coxiella burnetii*, an obligate intracellular bacterial pathogen. In the current manuscript, we identify and characterize a novel *Coxiella* secreted effector protein, *CbEPF1*, which facilitates membrane contact sites between the host ER and lipid droplets. To our knowledge, this is the first example of a bacterial secreted protein which manipulates inter-organelle communication away from the BCV.

We thank all three reviewers for their careful consideration and constructive criticisms of this manuscript. Overall, the reviews were very positive and acknowledged the impact of our findings that will appeal to cell biologists as well as the researchers in the infection and immunity communities.

Reviewers 1 and 3 requested functional relevance to *Coxiella* infection. We are performing additional experiments to examine the presence of ER-lipid droplet contacts in *Coxiella*-infected cells. **Reviewers 2 and 3** requested additional experiments regarding the amphipathic helix of *CbEPF1* and how *CbEPF1* localizes to the lipid droplets. We are performing new experiments using *CbEPF1* domain mutants as well as a mutant in the amphipathic helix. A point-by-point response to all comments is included below at the end of this document.

2. Description of the planned revisions

1. All three reviewers inquired about the role of *CbEPF1* during *Coxiella* infection. Thus far we have been unable to generate an *CbEPF1* mutant in *Coxiella*, and to our knowledge there is no *CbEPF1* mutant available in other labs. *CbEPF1* is a novel protein with no identifiable protein homologs; thus, there are no antibodies available to detect endogenous protein. To attempt to link *CbEPF1*-induced ER-LD contacts to *Coxiella* infection, we are conducting electron microscopy experiments of *Coxiella*-infected macrophages and will determine whether infected cells have increased ER-LD contacts compared to uninfected cells.
2. An additional question was how *EPF1* targets localization to LDs. *CbEPF1* contains an amphipathic helix, which is responsible for targeting some mammalian proteins to LDs. However, mutating the *CbEPF1* amphipathic helix had no effect on localization. For the revision, we are generating a series of *CbEPF1* domain mutants to determine which portion of *CbEPF1* is necessary and sufficient for LD localization. This data will be included in the revised manuscript, along with the amphipathic helix data.

3. The *CbEPPF1* FFAT motifs are responsible for binding VAPB and MOSPD2, both members of the VAP family of ER proteins. For the revision, we will test whether a third member of this family, VAPA, also binds to one or both of the *CbEPPF1* FFAT motifs. This will be done using pulldowns of both wildtype and mutant *CbEPPF1*-GFP, followed by immunoblotting for VAPA.
4. As a control experiment to show that *CbEPPF1*-induced ER-LD contacts are not a result of ectopic expression of KDEL-BFP, we plan to stain endogenous VAPB along with lipid droplets in *CbEPPF1*-GFP-expressing cells.
5. We are performing additional experiments to increase experimental power for Figure 5a, which examines the number of lipid droplets in *CbEPPF1*-GFP-expressing cells.

3. Description of the revisions that have already been incorporated in the transferred manuscript

The following text revisions have been completed and summarized below. Reviewer comments are in purple text and italicized; author responses are in black text.

1. Questions regarding the LD clustering phenotype observed with the *CbEPPF1* F3mut:

A CbEPPF1 FFAT motif mutant (F3mt) associates with LD that do not associate with the ER marker KDEL, and causes LD clustering (Fig 4). The authors speculate that the lack of LD-ER interaction results in LD-LD interaction, potentially via interaction of unidentified protein(s) on the LD surface. What is the biological relevance of the LD clustering phenotype? What is known about the role of LD clustering vs ER-LD contact, in the context of lipid metabolism?

We have extended our discussion on LD clustering phenotype and its role in lipid metabolism in the revised manuscript, Lines 242-257

"clustering" LDs w F3mt: could this indicate dimer formation by CbEPPF1?

While we have no evidence thus far, we cannot rule out *CbEPPF1* dimers. This possibility is now addressed in the manuscript, lines 197-199.

LD proximities to each other dominate. It's not 100% clear that LDs cluster as their proximities are not universal: "LD-LD interactions" may be (very) weak.

We agree with the reviewer that, at this point, we cannot determine that the LDs are directly interacting with each other. We have modified the text to reflect this, lines 193-199

Line 226: no sequence homology: misses the point- there is the common feature of an AH

Revision Plan

We have revised this statement and paragraph to better emphasize the functional homology with known eukaryotic proteins, in the absence of sequence homology (lines 260-285).

2. Representative images for LD size do not reflect quantitative data:

CbEPF1-GFP overexpression result in an increase in the number of LD per cell independently of the FFAT motifs (Fig 5A). CbEPF1-GFP overexpression also result in an increase in LD diameter; however, this phenotype is dependent on wild-type FFAT motifs (Fig 5B).

Quantification and corresponding statistical analysis support these conclusions. However, the representative images are not necessary in line with the bar graphs. For example, there appear to be less LD upon expression of F1mt or F2mt, compared to WT. Additionally, the increase in size is moderate and hard to appreciate in the IF images.

We increased the magnification to make the LD sizes more clear and replaced representative images to better reflect the quantitated data (Fig. 5C).

3. Questions regarding the relevance of LDs during *Coxiella* infection (multiple comments)

*It is also unclear, if/how an increase in LD number and/or size is biologically relevant in the context of *Coxiella* infection.*

*The present study suggest that LD may be important for *Coxiella* intracellular life cycle. Are LD formation induced upon infection?*

*One may also expect that inhibition of LD formation would affect bacterial replication and that stimulation may promote growth/vacuole maturation? Any experiments addressing the biological relevance of the present findings in the context of *Coxiella* infection would tremendously increase the impact of the study.*

A previous study from our lab determined that one or more *Coxiella* T4BSS manipulates LD metabolism and leads to more LDs in infected macrophages (Mulye et al, PMID 29390006). Based on the data presented in the current manuscript, we believe that *CbEPF1* is one of these *Coxiella* effector proteins. The current findings are placed into this context in lines 286-309.

4. Interaction between *CbEPF1* and VAPs

*Issue to be discussed, as probably too difficult to experiment on: when *EPF1* is on the ER does it engage *vap* only weakly (implying a means to mask its motifs), since if the interaction is strong *vap* is then unable to bind other partners?*

We agree that the strength of *EPF1*-VAP binding, and how it may change during the various stages of LD metabolism, would be very challenging to determine experimentally. We have added this possibility to the discussion (lines 231-241):

Revision Plan

“While our findings demonstrate that CbEPF1 interacts with VAPs on the ER through its FFAT motifs when localized on LDs, we currently lack evidence regarding its interaction with VAPs while localized on the ER itself. Given that the ER is characterized by a double-membrane structure, distinct from the monolayer structure of the LD surface, we hypothesize that CbEPF1 may undergo conformational changes as it shuttles between the ER and the LD surface. Consequently, CbEPF1 could potentially adopt a structural form that allows it to interact with VAPs solely when localized on LDs.”

5. Minor changes and clarifications

Fig 2B: add more arrowheads/arrows to fit legend (says they are both multiple)

Additional arrows have been added to Fig 2B.

FFAT selectivity for MOSPD2: say if this fits the di Mattia or (as appears likely) it extends the known differences between VAPA/B and MOSPD2.

This has been added to the discussion, lines 175-179

Also, the (admittedly low) level of ER targeting is possibly slightly reduced by F3mt, as shown by greater GFP in the nucleus in the single cells shown. If this is a feature of the whole field of cells, it implies that the FFATs normally work with the AHs to target EPF1 to the ER.

We examined all of our F3mt images and found none localized to the host cell nucleus. Upon further examination of the image in Supplemental Figure 1, we realized that we inadvertently chose a focal plane which is just above the nucleus and the “nuclear localization” is in fact the ER network. The revised figure (now Supplemental Figure 2) is of the same field as the original manuscript, but a focal plane going through the nucleus is now shown.

Line 245: “MOSPD2, a sole VAP that is known to localize on LD surfaces” (worth citing Zouiouich again here).

This citation has been added (line 281-282)

Line 259/260: this “suggestion” about cholesterol should be toned down. It is a speculation that could be tested in future, but the data here do not suggest it.

This possibility has been re-phrased as a hypothesis for future experiments (lines 306-309).

Discuss whether it is 100% certain that EPF1 is in the host cytosol or whether some experiment(s) at a future date (proteomic/western blotting) will be needed to make that conclusion 100% secure.

We have expanded our discussion of EPF1 localization (lines 231-241)

Revision Plan

Authors must perform colocalization analyses to substantiate the claims for ER/LD localization. The authors refer to "extended ER-LD contacts" in figure 2B and the text. These data need to be supported with colocalization analysis between BFP-KDEL and the GFP channel.

Colocalization between BFP-KDEL and GFP has been added to as a Supplemental Figure 1.

Related to Figure 4B: Please show the LD phenotype of untransfected, and CbEPF1-GFP transfected cells also. Can the authors provide a means to quantify the clustering of LDs.

Control images for GFP and CbEPF1-GFP have been added to Fig 4C of the revised manuscript, while the LD phenotype of CbEPF1-F1mt-GFP and CbEPF1-F2mt-GFP shown in Supplemental Figure 3.

In regard to quantitation of LD clustering, we have not devised a method that would provide new information and feel that the images clearly demonstrate the phenotype.

Figure 5A and B. It is not clear from the figure legend whether the data are pooled from multiple experiments or a single experiment. Experimental replicates must be incorporated in the final analysis.

Additional experimental details have been included in the revised manuscript, lines 514-522.

Line 36: "maintain" instead of "maintains"

This typo has been corrected.

6. Description of analyses that authors prefer not to carry out

Please include a point-by-point response explaining why some of the requested data or additional analyses might not be necessary or cannot be provided within the scope of a revision. This can be due to time or resource limitations or in case of disagreement about the necessity of such additional data given the scope of the study. Please leave empty if not applicable.

- 1. Endogenous localization of CbEPF1.** We attempted to localize endogenous CbEPF1 by expressing a tagged version in *Coxiella*. However, we were unable to visualize the protein in infected cells despite multiple approaches. This is most likely due to low expression level, which is a common feature of bacterial effector proteins. In fact, to our knowledge, there have been no reports of endogenous *Coxiella* effector proteins being visualized in the host cell cytoplasm. We will continue to address this important question as new tools become available.
- 2. Compare levels of ectopic expression to endogenous expression.** It would be very difficult to compare CbEPF1 levels between infected cells and ectopic expression, as we do not have an antibody against CbEPF1. While qPCR is feasible, regulation of T4BSS secretion of effector proteins is poorly understood and we would not be able to determine how much of the protein is in the bacteria versus in the host cell cytoplasm.

Revision Plan

- 3. Is VAP required for *Coxiella* infection?** As reviewer 2 pointed out in the cross-cutting comments, VAP binds hundreds of proteins, including some that localize to lipid droplets. Further, as *CbEPF1* binds VAPB and MOSPD (and most likely VAPA), we would need to knockdown all three proteins, which would make these experiments difficult to interpret.
- 4. Isolate lipid droplets to demonstrate that *CbEPF1* is present.** We believe we have presented clear evidence that *CbEPF1* is found on lipid droplets. Given this, and the amount of time and resources to obtain a pure lipid droplet prep from *CbEPF1*-expressing cells, we do not feel this request is necessary.

Dear Stacey,

Thank you for the transfer of your research manuscript to our journal. As discussed, we would like to invite you to revise your study according to your submitted revision plan. Evidence for an effect on Coxiella replication and infection is not required. Referee #2 brought forward strong arguments against the need for such experiments and we agree with that. All other points seem feasible, I feel. Lipidomics and proteomics of LDs is something for the future.

Please address all referee concerns in a complete point-by-point response. Your revised manuscript will be sent back to the same reviewers who reviewed your manuscript at Review Commons and acceptance of the manuscript will depend on a positive outcome of this second round of review. It is EMBO Reports policy to allow a single round of revision only and acceptance or rejection of the manuscript will therefore depend on the completeness of your responses included in the next, final version of the manuscript.

We realize that it is difficult to revise to a specific deadline. In the interest of protecting the conceptual advance provided by the work, we recommend a revision within 3 months (July 4th). You already indicated that this timeframe is reasonable, but please let me know closer to this date if you require more time to complete the revisions.

*******IMPORTANT NOTE:**

We perform an initial quality control of all revised manuscripts before re-review. Your manuscript will FAIL this control and the handling will be delayed IN CASE the following APPLIES:

- 1) A data availability section providing access to data deposited in public databases is missing. If you have not deposited any data, please add a sentence to the data availability section that explains that.
- 2) Your manuscript contains statistics and error bars based on $n=2$. Please use scatter blots in these cases. No statistics should be calculated if $n=2$.

When submitting your revised manuscript, please carefully review the instructions that follow below. Failure to include requested items will delay the evaluation of your revision. *****

- 1) a .docx formatted version of the manuscript text (including legends for main figures, EV figures and tables). Please make sure that the changes are highlighted to be clearly visible.
- 2) individual production quality figure files as .eps, .tif, .jpg (one file per figure). Please download our Figure Preparation Guidelines (figure preparation pdf) from our Author Guidelines pages <https://www.embopress.org/page/journal/14693178/authorguide> for more info on how to prepare your figures.
- 3) a .docx formatted letter INCLUDING the reviewers' reports and your detailed point-by-point responses to their comments. As part of the EMBO Press transparent editorial process, the point-by-point response is part of the Review Process File (RPF), which will be published alongside your paper.
- 4) a complete author checklist, which you can download from our author guidelines (). Please insert information in the checklist that is also reflected in the manuscript. The completed author checklist will also be part of the RPF.
- 5) Please note that all corresponding authors are required to supply an ORCID ID for their name upon submission of a revised manuscript (). Please find instructions on how to link your ORCID ID to your account in our manuscript tracking system in our Author guidelines ()
- 6) We replaced Supplementary Information with Expanded View (EV) Figures and Tables that are collapsible/expandable online. A maximum of 5 EV Figures can be typeset. EV Figures should be cited as "Figure EV1, Figure EV2" etc... in the text and their

respective legends should be included in the main text after the legends of regular figures.

7) Please note that a Data Availability section at the end of Materials and Methods is now mandatory. In case you have no data that requires deposition in a public database, please state so instead of referring to the database.

See also < <https://www.embopress.org/page/journal/14693178/authorguide#dataavailability>>).

Please note that the Data Availability Section is restricted to new primary data that are part of this study and should not contain other statements about data or reagent availability upon request.

Additional information on source data and instruction on how to label the files are available .

10) Figure legends and data quantification:

- the name of the statistical test used to generate error bars and P values,
- the number (n) of independent experiments (please specify technical or biological replicates) underlying each data point,
- the nature of the bars and error bars (s.d., s.e.m.)

- If the data are obtained from n {less than or equal to} 5, show the individual data points in addition to the SD or SEM.

- If the data are obtained from n {less than or equal to} 2, use scatter blots showing the individual data points.

11) Our journal encourages inclusion of *data citations in the reference list* to directly cite datasets that were re-used and obtained from public databases. Data citations in the article text are distinct from normal bibliographical citations and should directly link to the database records from which the data can be accessed. In the main text, data citations are formatted as follows: "Data ref: Smith et al, 2001" or "Data ref: NCBI Sequence Read Archive PRJNA342805, 2017". In the Reference list, data citations must be labeled with "[DATASET]". A data reference must provide the database name, accession number/identifiers and a resolvable link to the landing page from which the data can be accessed at the end of the reference. Further instructions are available at .

12) All Materials and Methods need to be described in the main text. We would encourage you to use 'Structured Methods', our new Methods format. According to this format, the Methods section should include a Reagents and Tools Table (listing key reagents, experimental models, software and relevant equipment and including their sources and relevant identifiers) followed by a Methods and Protocols section in which we encourage the authors to describe their methods using a step-by-step protocol format with bullet points, to facilitate the adoption of the methodologies across labs. More information on how to adhere to this format as well as downloadable templates (.doc or .xls) for the Reagents and Tools Table can be found in our author guidelines: < <https://www.embopress.org/page/journal/14693178/authorguide#manuscriptpreparation>>.

An example of a Method paper with Structured Methods can be found here: .

13) As part of the EMBO publication's Transparent Editorial Process, EMBO Reports publishes online a Review Process File to

accompany accepted manuscripts. This File will be published in conjunction with your paper and will include the referee reports, your point-by-point response and all pertinent correspondence relating to the manuscript.

Kind regards,

Martina

We sincerely thank the reviewers for their constructive criticism, suggestions, and editorial suggestions. We have addressed their comments, as detailed below and highlighted in the revised manuscript.

Reviewer #1 (Evidence, reproducibility and clarity (Required)):

Summary:

The authors report a Coxiella burnetii effector protein, CbEPF1, which associate with lipid droplets (LD) at points of contact with the endoplasmic reticulum (ER). The presence of two FFAT motifs in CbEPF1 mediates CbEPF1 interaction with the ER protein VAP, LD-ER interaction, and an increase in LD size. Based on these results the authors put forward a model by which translocation of CbEPF1 into the host cell cytosol results in regulation of host cell lipid metabolism via the formation of ER-LD contact. The study relies heavily on fluorescence microscopy of ectopic (over)expression of CbEPF1 in eukaryotic cells.

Major comments:

CbEPF1 and ER-LD contact:

The immunofluorescence analysis of cell ectopically expressing CbEPF1-GFP provides convincing evidence that CbEPF1-GFP associates with LD and that CbEPF1-positive LD are positive for ER markers such as BFP-KDEL and VAPB (Fig 1&2). However, the study would benefit from looking at endogenous proteins to rule out any potential overexpression artefacts. Does endogenous VAP localize to CbEPF1-positive LD?

Using immunofluorescence for endogenous VAPA and VAPB in cells expressing CbEPF1-GFP, we showed both VAPA and VAPB localize to CbEPF1-positive LDs (Fig. 3D and 3E). We were unable to detect endogenous MOSPD2 with available antibodies; consequently, we used ectopic expression of GFP-MOSPD2 with mCherry-CbEPF1 (Fig. 3F). The VAPB ectopic expression data from the original submission has been removed and replaced with endogenous data.

Does CbEPF1 expressed from Coxiella (endogenous protein, or at least a tagged protein expressed from the bacteria) localize to LD?

Localizing endogenous secreted bacterial proteins is very challenging and, to our knowledge, has not been achieved in *Coxiella*. This could be due to low expression levels or regulated secretion by the *Coxiella* Type 4B Secretion System. We do not have CBU1370/CbEPF1-specific antibody to detect endogenous protein expression.

While the immunofluorescence images are of very high quality and convincing, electron microscopy is necessary to ascertain that membrane contacts between LD and the ER are induced upon CbEPF1 overexpression.

We believe, along with Reviewer 2, that the confocal images are convincing evidence of ER-LD contacts, especially when taking into account the changes in localization between wildtype and CbEPF1 FFAT mutants. While we agree that electron microscopy may be beneficial, we argue that they will not provide additional insight or evidence.

CbEPF1 interaction with the ER is linked to the FFAT motifs (see below); what is known about CbEPF1 association with LD? Can a specific CbEPF1 domain be identified? In other words, does CbEPF1 contains 2 distinct membrane targeting domains that confer specificity to each of the contacting organelle (ER and LD in the present case) and thereby resemble other contact site localizing proteins.

Regarding domains: The majority of *Coxiella* secreted effector proteins are classified as hypothetical proteins with little to no homology to known proteins, although they may contain eukaryotic motifs or conserved domains. In the case of CbEPF1, we were unable to identify significant homology to any bacterial or eukaryotic proteins, and structure prediction tools (I-tasser and alpha fold) failed to generate a predicted structure with confidence. There is a predicted amphipathic helix in the C-terminus; amphipathic helices can be involved in localization to LDs. Interestingly, mutations in the amphipathic helix did not completely prevent CbEPF1 from localizing to LDs. Thus, to identify which portion of CbEPF1 is required for LD localization, we generated

truncated mutants of CbEPF1. We found that the 111 amino acids (101-213aa) present in the middle of CbEPF1 also target the truncated protein to LDs. Our sequence analysis revealed the central domain is a hydrophobic rich region on Kyte-doolittle scale (see figure below). While the C-terminal amphipathic helix and middle hydrophobic regions are sufficient for truncated protein localization to LDs, we believe both domains are crucial for full length CbEPF1 conformation and localization to LDs. This new data is presented in the revised manuscript (Fig. 6).

Figure: Hydropathy analysis from ProtScale using Hydropath/Kyte & Doolittle scale. The middle 111 amino acids that localize to LDs are boxed.

CbEPF1 FFAT motifs and VAP binding:

The similarity of the sequence of the CbEPF1 FFAT motifs to the canonical sequence (Fig 1A), combined with data with alanine substitution mutation of the essential residue in position 2 of the FFAT motifs (Fig 3, 4), strongly support that CbEPF1 contains 2 functional FFAT motifs that confer VAP binding.

FFAT motifs can mediate binding to VAPA, VAPB, and/or MOSPD2. Moreover, in addition to forming homodimer, VAPA, VAPB, and MOSPD2 can form heterodimers. What is the rationale for using VAPB (Fig 3CDE)?

We chose to begin with VAPB for practical reasons, as the VAPB expression constructs and VAPB antibodies were already available in our lab. We provide data for VAPA and MOSPD2 in the revised manuscript (3D, E, F, G, and H).

Do VAPA and/or MOSPD2 also yield positive results using the assays performed in Fig 3CDE, or is the CbEPF1-VAP interaction specific to VAPB?

CbEPF1 also interacts with VAPA (Fig. 3H) and MOSPD2 (Fig. 3G) in addition to VAPB.

In the same line, in Fig 3E, is it possible that the CbEPF1-MOSPD2 is indirect and due to VAP-MOSPD2 interaction?

As the mutation in the second FFAT motif (CbEPF1-F2 mutant) specifically hampered interaction with MOSPD2 without impacting interaction with VAPB, we believe CbEPF1-MOSPD2 interaction is direct and requires the second FFAT motif. In other words, the CbEPF1-MOSPD2 interaction does not depend on other VAPs (Fig. 3G).

Regarding the two FFAT motifs: are the two FFAT motifs redundant or synergistic? Although the data is not quantified, Figure 3E suggest a synergistic effect for VAP binding, however most IF data suggest redundancy. On the other end, the second FFAT motif seems necessary for MOSPD2 binding. Overall, clarifying CbEPF1

binding specificity towards a VAP/MOSPD2 and the role of each FFAT motif in this process could elevate the study by providing mechanistic insights into the hijacking of ER-LD contact sites by Coxiella and comparing and contrasting with the formation of ER-LD in naïve cells and/or the hijacking of VAP-dependent contact by other microbial pathogens.

Using individual FFAT motif mutants and pulldown experiments, we demonstrated that the second CbEPF1 FFAT motif is required for binding to MOSPD2 (Figure 3G). Based on microscopy, only one motif is required for required for CbEPF1-mediated ER-LD contacts (Fig.4A and 4B). Thus, for ER-LD contact formation, the data indicate that the motifs are redundant. Since VAPs dimerize (homodimers and heterodimers) on endoplasmic reticulum, presence of two FFAT motifs in CbEPF1 may be an evolutionary adaptation to enhance the binding with VAPs. Future experiments outside the scope of this manuscript include structural elucidation of CbEPF1 binding to the VAP protein family.

CbEPF1-mediated LD clustering in the absence of VAP binding

A CbEPF1 FFAT motif mutant (F3mt) associates with LD that do not associate with the ER marker KDEL, and causes LD clustering (Fig 4). The authors speculate that the lack of LD-ER interaction results in LD-LD interaction, potentially via interaction of unidentified protein(s) on the LD surface.

What is the biological relevance of the LD clustering phenotype? What is known about the role of LD clustering vs ER-LD contact, in the context of lipid metabolism?

We have extended our discussion on LD clustering phenotype and its role in lipid metabolism in the revised manuscript, Lines 292-307.

Could mechanistic characterization of this phenomenon provide insights in LD biology and/or the role of CbEPF1/ER-LD contacts in the context of Coxiella infection?

We agree with the reviewer that understanding role of CbEPF1-mediated ER-LD contacts in *Coxiella* infection would be a great value. However, due to lack of CbEPF1 mutants, we were unable to address this in our current manuscript.

CbEPF1-mediated increase in LD number and size

CbEPF1-GFP overexpression result in an increase in the number of LD per cell independently of the FFAT motifs (Fig 5A). CbEPF1-GFP overexpression also result in an increase in LD diameter; however, this phenotype is dependent on wild-type FFAT motifs (Fig 5B).

Quantification and corresponding statistical analysis support these conclusions. However, the representative images are not necessary in line with the bar graphs. For example, there appear to be less LD upon expression of F1mt or F2mt, compared to WT. Additionally, the increase in size is moderate and hard to appreciate in the IF images.

We increased the magnification to make the LD sizes more clear and replaced representative images (Fig. 7C) to better reflect the quantified data.

It is also unclear, if/how an increase in LD number and/or size is biologically relevant in the context of Coxiella infection.

A previous study from our lab determined that one or more *Coxiella* T4BSS manipulates LD metabolism and leads to more LDs in infected macrophages (Mulye et al, PMID 29390006). Based on the data presented in the current manuscript, we believe that CbEPF1 is one of these *Coxiella* effector proteins. The current findings are placed into this context in lines –351-362.

Regarding potential mechanism(s), it is also unclear how CbEPF1 is promoting an increase in LD number. De novo LD production at the ER is unlikely given that the FFAT motifs, and therefore ER association, are not required. What would an alternative mechanism be, and can it be experimentally tested?

CbEPF1 is enriched at LD biogenesis sites before translocating to matured LD surface (Fig. 1C). Based on data that the FFAT motifs are not involved in increasing LD number, we hypothesize that CbEPF1 may have dual functions. First, it may promote LD biogenesis in an FFAT-independent process. Once CbEPF1 has moved to the mature LD, it then promotes ER-LD contacts in an FFAT-dependent manner. Testing this hypothesis, which we plan in future studies, involves further characterization of the N-terminal domain, identifying CbEPF1-interacting proteins at different stages of LD biogenesis, and advanced live cell imaging.

Regarding the increase in LD size, the author suggest that the phenotype could be due to impaired lipid transfer from the ER to LD. This is an interesting model. Do the authors envision that CbEPF1 is a lipid transfer protein and/or act on ER-LD associated lipid transfer? Can either be experimentally tested? Altogether, strengthening this aspect of the study would clarify the proposed model and significantly increase the impact of the study.

There are several examples of FFAT motif-containing proteins which function as a lipid transfer protein (OSBP, PMID: 24209621; CERT, PMID: 16895911; ORP1L, PMID: 31570833; STARD3, PMID: 28377464; VPS13C, PMID: 30093493). While we have not identified a conserved domain or structure in CbEPF1, it is possible that CbEPF1 does have lipid transfer ability. While this requires extensive experimentation outside the scope of this paper, we plan to test this exciting hypothesis in the near future.

Functional relevance:

One aspect that is not addressed in the study, is what are the benefit(s), if any, of CbEPF1 translocation into the host cytosol and targeting to ER-LD contact?

This could be addressed by assessing the phenotype of a Coxiella CbEPF1 mutant?

We agree with the reviewer that this is an important question to address. Unfortunately, we do not currently have a *Coxiella* CbEPF1 mutants to address this question. We are actively working to generate knockdown or knockout strains. However, as genetic manipulation of *Coxiella* is challenging and even in the best of circumstances it takes several months to generate and validate mutants due to the slow growth phenotype (doubling time 10-12 hours), we cannot address this question in the current manuscript.

On the flip side, is VAP required for Coxiella intracellular growth/vacuole maturation?

In another project in the lab, we found that siRNA knockdown of individual VAPs (VAPB and VAPA) did not have a significant impact on *Coxiella* growth and vacuole. Based on overlapping or functional redundancy, it may be necessary to target multiple VAPs. However, as reviewer 2 indicated, VAPs have hundreds of interacting partners and majority associated with membrane contact site formation and lipid transfer at inter-organelle contact sites. Therefore, a phenotype from VAP-deficient cells may be difficult to interpret.

Another avenue would be to investigate if CbEPF1 affects the lipid composition of the CCV.

We agree that it would be very informative to determine the lipid composition of the CCV. Unfortunately, CCV purification has not yet been achieved in the field.

The present study suggest that LD may be important for Coxiella intracellular life cycle. Are LD formation induced upon infection?

Our previous studies demonstrated that *Coxiella* induces LDs in macrophages (Mulye et al, 2018 PMID 29390006). This is included in the text, lines –351-352.

Are ER-LD contact increased upon infection?

We analyzed ER-LD contacts in *Coxiella*-infected macrophages using transmission electron microscopy (TEM). We found *Coxiella* infection significantly increases ER-LD contacts in the host cells compared to the uninfected cells; this data is presented as Fig. 5A and B of the revised manuscript.

One may also expect that inhibition of LD formation would affect bacterial replication and that stimulation may promote growth/vacuole maturation? Any experiments addressing the biological relevance of the present findings in the context of Coxiella infection would tremendously increase the impact of the study.

We examine the impact of LD formation and catabolism on *Coxiella* infection in our previous study (Mulye et al, 2018, PMID 29390006). We have extended our discussion section to emphasize the significance of LDs during *Coxiella* infection (lines 351-373).

Minor comments:

None. The manuscript is well written and easy to follow.

Reviewer #1 (Significance (Required)):

Inter-organelle communication through the formation of membrane contact between two apposing organelles is critical to maintain host cell homeostasis via the transfer of small molecules such as lipid and ions. Bacterial and viral pathogens have been shown to manipulate proteins that localize to cellular membrane contact and to promote membrane contact between their intracellular vacuole and the ER. Both viral and bacterial proteins that contains FFAT motifs and interact with VAP have been described in the context of tethering of the ER to the membrane compartment in which the pathogen replicate.

The present study stands out by the characterization of a FFAT motifs-containing bacterial effector protein that targets cellular contact, ER-LD more specifically. The significance is 2-fold. It expands the list of FFAT-motif containing proteins in pathogens, reinforcing the idea that VAP/membrane contact may be a universal mechanism of host-pathogen interaction, which will be of interest to those studying host-microbe interaction in basic or translational research settings. The study also has the potential to move forward research on LD biology and LD-ER contact, which will be of interest to cell biologists in general, and to the evergrowing membrane contact community, specifically.

Reviewer #2 (Evidence, reproducibility and clarity (Required)):

This is a strong manuscript about the existence of proteins coded by intracellular parasites (here Coxiella) that have evolved to parasitise the lipid transport machinery of their hosts. This is a first in that the parasite protein acts at a distance from the parasite itself, manipulating two of the host organelles - and not acting at their site of contact with PVs. There is considerable research into one protein and its effect when expressed by itself.

Despite all the advances there are a couple of areas where the manuscript can be improved, and a few extra fairly straightforward experiments added about the amphipathic helix. Even though these are unlikely change the overall message, they would make the story more complete.

MAJOR POINTS

More details are required about the amphipathic helix. Check that the AH does target LDs by expression of the AH alone in a GFP chimera +/- oleate and then mutagenesis. Also show the AH in a helical wheel projection (eg by Heliquest) and say if it aligns with similar AHs in homologs (see my point below).

We investigated the requirement of amphipathic helix for CbEPF1 localization to LDs. While the AH (AH-GFP) localized to LDs (Fig. 6D) and mutations in AH (AHmt-GFP) disrupted its localization to LDs (Fig. 6E), mutations in AH did not completely prevent CbEPF1 localization to LDs (Fig. 6C). Therefore, to identify whether there are additional factors responsible for CbEPF1 localization to LDs, we generated truncated mutants of CbEPF1. In our analysis we found that the 111 amino acids present in the middle of CbEPF1 also target the truncated protein to LDs. Our sequence analysis revealed the central domain is a hydrophobic rich region (see figure below). While the C-terminal amphipathic helix and middle hydrophobic regions are sufficient for truncated protein localization to LDs, we believe both are crucial for full length protein conformation and localization to LDs. We have provided the truncated protein analysis data in the revised manuscript (Fig. 6).

Figure: Hydropathy analysis from ProtScale using Hydropath/Kyte & Doolittle scale. The middle 111 amino acids that localize to LDs are boxed.

Fig 1B: In infected cells, do the affected LDs tend to be close to the PVs?

We do not consistently observed LDs in close proximity to CCVs.

Also in Fig 1B: highlight small KDEL+ve ER rings around LDs here.

We appreciate the suggestion. However, we intentionally avoided highlighting ER association (KDEL+ve ER rings around LDs) with LDs in this section, as we wanted the reader to focus on CbEPF1-GFP localization in the infected cells.

Study whether LDs have these in infected cells without the confounder (?artefact) of EPF1 over-expression

We analyzed ER-LD contacts in *Coxiella*-infected macrophages using transmission electron microscopy (TEM). We found *Coxiella* infection significantly increases ER-LD contacts in the host cells compared to the uninfected cells; this data is presented as Fig. 5A and B of the revised manuscript.

Fig 2A the ER looks quite different here from Fig 1B, even at t0. Grossly the strands are spaced wider apart. In detail there are no rings around LDs. Can the authors explain this? Which morphology is common, especially in cells early in infection without co-expressed protein?

We performed Fig 2A experiment in uninfected cells while Fig 1B in *Coxiella*-infected cells. The ER morphology (strand spacing and morphology) could be different due to a stress response due to the infection. Further, other *Coxiella* T4BSS effectors are known to target the ER (PMID: 33355405; PMID: 25605765).

Regarding ER rings around LDs: In general, the number of LDs in uninfected HeLa cells is very low (3-10 LDs/cell; Fig 5A). We observed ER rings around large/matured LDs where CbEPF1-GFP also localized as ring, but not around LDs at their early biogenesis state. The difference in ER rings between Fig 2A (early time points of oleate treatment) and Fig 1B could be due to the maturation stage of LDs. For example, in Fig 2A CbEPF1-GFP is localized in the ER and the LDs are relatively smaller, hence ER did not show wrapping/ring around LDs. Future studies will address this question.

Fig 6 & Line 237: "As the N-terminal region of CbEPF1 is undefined": I suggest that the authors could do more here. At minimum change model to highlight the strong probability that the N term is a globular domain that functions at the LD ER interface. (What are three other unidentified LD proteins? I suggest omitting them).

We are unsure of what the reviewer is referring to in the "three other unidentified LD proteins."

The discussion has been revised to reflect the possibility of a globular domain (Line 335-340). We agree with the reviewer future experiments to define the N-terminal region will help us understand the function of CbEPF1 at ER-LD contact sites. We attempted to predict structure and find a globular domain using models from alphafold, I-tasser, and Raptorx; none of the predicted models showed significant confidence score.

Although the Alphafold prediction for EPF1 is low confidence only, in a few minutes of BLAST searching I found the homolog A0A1J8NR10_9COXI (also FFAT+ve) which has a moderately confident structural prediction for its N terminus. This model has a quite large internal hydrophobic cavity, indicating lipid transfer capability and function similar to known LTPs. This means that action as a "tether" possibly results from experiments with viral promoters (see minor point on terminology).

We appreciate the reviewer's insight and suggestion of CbEPF1 containing an N-terminal hydrophobic cavity which could serve as an LTP. We plan to address this possibility in future studies, including structural determination of CbEPF1.

Minor:

Fig 2B: add more arrowheads/arrows to fit legend (says they are both multiple)

Additional arrows have been added to Fig 2B.

FFAT selectivity for MOSPD2: say if this fits the di Mattia or (as appears likely) it extends the known differences between VAPA/B and MOSPD2.

This has been added to the discussion, lines 180-184

Also say if VAPA is expected to behave as VAPB

In the revised manuscript, we tested VAPA and found VAPA interacts with CbEPF1 similar to VAPB (Figure 3A and 3H).

Explain how "Mutations in the CbEPF1 FFAT motif(s) did not influence CbEPF1-GFP localization to either the host ER (Supplementary Fig. 1)". In F3mt this shows that EPF1 has a way to target ER other than FFAT/VAP. Discuss if that is via AH insertion in ER.

Our data suggest that CbEPF1 is initially found in the ER in an FFAT-independent manner. Once CbEPF1 has translocated to LDs, the FFAT motifs then bind VAPs on the ER, mediate ER-LD contacts. Thus, we do not expect either single or double FFAT mutations to alter the ER localization. FFAT motifs are required for proteins to interact with VAPs in the ER. No previous studies have shown FFAT motifs are responsible for protein localization per se to ER.

Also, the (admittedly low) level of ER targeting is possibly slightly reduced by F3mt, as shown by greater GFP in the nucleus in the single cells shown. If this is a feature of the whole field of cells, it implies that the FFATs normally work with the AHs to target EPF1 to the ER.

We examined all of our F3mt images and found none localized to the host cell nucleus. Upon further examination of the image in Supplemental Figure 1, we realized that we inadvertently chose a focal plane which is just above the nucleus and the "nuclear localization" is in fact the ER network. The revised Figure EV2 is of the same field as the original manuscript, but a focal plane going through the nucleus is now shown.

"clustering" LDs w F3mt: could this indicate dimer formation by CbEPF1?

While we have no evidence thus far, we cannot rule out CbEPF1 dimers. This possibility is now addressed in the manuscript, lines 201-203.

Note: to me it appears wrong to describe fig 4A as showing ER exclusion.

In Fig 4A figure legends, we stated 'CbEPF1-F3mt-GFP failed to induce ER-LD contact sites'. Our interpretation of ER exclusion is from Fig 4B, where we performed 3D-rendering of Z-stack and observed the ER exclusion around LD clusters.

LD proximities to each other dominate. It's not 100% clear that LDs cluster as their proximities are not universal: "LD-LD interactions" may be (very) weak.

We agree with the reviewer that, at this point, we cannot determine that the LDs are interacting with each other. We have modified the text to reflect this, lines 199-202.

Fig 5: can levels of EPF1 here be compared to those in cells undergoing natural infection (approximate comparison by qPCR better than nothing if no antibodies are available)?

It would be very difficult to compare EPF1 levels between infected cells and ectopic expression, as we do not have an antibody against EPF1. While qPCR is feasible, regulation of T4BSS secretion of effector proteins is poorly understood and we would not be able to determine how much of the protein is in the bacteria versus in the host cell cytoplasm.

Fig 5a: would it be possible to increase the number of cells counted to attempt to make the reduced number of LD in F3mt significant?

We increased the number of counted cells, which led to statistical significance (Fig. 5A and 5B).

Minor

Line 226: no sequence homology: misses the point- there is the common feature of an AH

We have revised this statement and paragraph to better emphasize the functional homology with known eukaryotic proteins, in the absence of sequence homology (lines 322-324).

Issue to be discussed, as probably too difficult to experiment on: when EPF1 is on the ER does it engage vap only weakly (implying a means to mask its motifs), since if the interaction is strong vap is then unable to bind other partners?

We agree that the strength of EPF1-VAP, and how it may change during the various stages of LD metabolism, would be very challenging to determine experimentally. We have added this possibility to the discussion (lines 282-291):

Line 245: "MOSPD2, a sole VAP that is known to localize on LD surfaces" (worth citing Zouiouich again here).

This citation has been added (line 346)

Do the cells/tissues infected by Coxiella express MOSPD2?

Coxiella primarily infects alveolar macrophages during a natural infection. According to the human protein atlas, MOSPD2 protein expression is observed in multiple cell types, including alveolar macrophages.

Line 259/260: this "suggestion" about cholesterol should be toned down. It is a speculation that could be tested in future, but the data here do not suggest it.

This possibility has been re-phrased as a hypothesis for future experiments (lines 372-375).

'Tether' this word implies more than just bridging but also a role in the physical formation of the contact. Since EPF1 most likely has an LTP domain, it seems linguistically confusing to refer to it as a tether, especially since the experiments that physically later LD-ER contact involve probable over-expression.

While we acknowledge that ER wrapping around CbEPF1-localized LDs observed in overexpression conditions necessitates further investigation using endogenously tagged *C. burnetii* strains or CbEPF1-specific antibodies to assess its impact on ER-LD contacts under basal expression levels. However, based on our current observations of CbEPF1's interaction with VAPs and its ability to induce ER-LD contacts, we find the term 'molecular tether' appropriate. Several FFAT motif containing proteins that has lipid transfer domain and interacting with VAPs to establish inter-organelle contacts have been commonly referred as molecular tethers (e.g., ORP1L, PMID: 31570833; STARD3, PMID: 24105263, 28377464; VPS13C, PMID: 30093493; ACBD5, PMID: 35019937). Hence, we believe that referring CbEPF1 as a molecular tether is justified.

Discuss whether it is 100% certain that EPF1 is in the host cytosol or whether some experiment(s) at a future date (proteomic/western blotting) will be needed to make that conclusion 100% secure.

We have expanded our discussion of CbEPF1 localization (lines 281-291)

****Referees cross-commenting****

We appreciate reviewer 2 for the suggestions in the cross-comments, which significantly benefited our revision.

COMMENT 1

I realise that both reviewer 1 and reviewer 3 have considered this MS carefully, but I think that their reviews could be improved in some respects. I will add two comments, one for each of the other reviews.

Reviewer 1.

The review poses multiple questions to the authors suggesting that answering these questions experimentally would strengthen the paper. Some of the points seem to misunderstand what is the accepted standard for membrane cell biological research into membrane contact sites. While it might be that the authors can rebut these points, I think it is preferable to use Cross Commenting as an opportunity to address these issues beforehand.

Major Comment 1: CbEPF1 and ER-LD contact

Looking at endogenous proteins:

I wondered about the same point, but I concluded that this is not likely to be possible in the scope of this submission. If it were possible then I guess the authors would have attempted it. Looking on Google Scholar I could find no example of an endogenous Coxiella proteins being tagged in the bacterial genome. So the only way to find the portion is via an antibody. Assuming the authors do not have one, I do not think we should ask for one at this stage in the publication process.

Electron microscopy:

the reviewer is incorrect to say that this is necessary. It may be the gold standard, but it is a huge amount of extra work. Furthermore, it is not at all necessary when the protein in question localises clearly to the interface between organelles identified by confocal microscopy.

Can a specific CbEPF1 domain be identified?

Here a Amphipathic Helix has been identified, but the lack of dissection of that region by the authors explains this question by the reviewer, which is also shared by Reviewer 3. I agree with the implication that more should be done to dissect that.

Major Comment 2: CbEPF1 FFAT motifs and VAP binding

Are the two FFAT motifs redundant or synergistic?

I would say that the authors have addressed that to a reasonable extent

CbEPF1 binding specificity towards a VAP/MOSPD2

Ditto

Major Comment 3: LD clustering

Since this is an effect of mutated protein only, I think that the 3 questions posed at the end here need only be addressed in Discussion.

Major Comment 4: CbEPF1-mediated increase in LD number and size

less LD upon expression of F1mt or F2mt, compared to WT:

this seems wrong. The numbers are the same. The comment about IF images are unjustified as they have been quantified and do show a difference.

I agree that the biological relevance is unclear, and that this might be addressed. That would require making a mutant Coxiella strain. While that would make a big difference to this work, my feeling is that this is well over a year's work. I would be guided by the authors on that and I would not suggest it as required for this MS.

De novo LD production at the ER is unlikely:

This statement is ill-considered as the FFAT motifs ARE required (Fig 5). Furthermore, in all systems ever reported de novo LD production takes place at the ER, so any alternative would be quite extraordinary.

Altogether, strengthening this aspect of the study:

In my view, this area does not need more work and it would not be constructive to ask for more.

Major Comment 5: Functional relevance

assessing the phenotype of a Coxiella CbEPF1 mutant

I agree that this would be good, but it mightn't be feasible within the confines of this one paper. In the various projects that have made transposon mutants of Coxiella, has a strain been made that affects EPF1? If not, then the authors should state this and discuss it as work for the future. The reviewers cannot expect any experiments!

Is VAP required for Coxiella intracellular growth/vacuole maturation?

On the surface this suggestion seems to offer an experimental route to understanding EPF1. However, VAP binds to >100 cellular proteins, many relating to lipids traffic and a considerable number of these already localised to lipids droplets (ORP2, MIGA2, VPS13A/C). It is therefore unlikely that such an experiment would be interpretable, and I recommend that this request be reconsidered.

Are LD formation induced upon infection? Are ER-LD contact increased upon infection?

These are very reasonable ideas and the results would be interesting additions to this paper.

COMMENT 2

I have given one set of comments already. Here are my comments for Reviewer 3.

The review makes a few assumptions that I question. While it might be that the authors can rebut these assumptions, I think it is preferable to use Cross Commenting as an opportunity to address these issues

beforehand.

Major Point 1:

What is surprising is that the BFP-KDEL signal is also localizing to the LD surface:

"Surprising" is misguided, as it seems to deny the probability that there is a class of proteins that sit at organelle interfaces binding to both partners simultaneously. Maybe the reviewer means "significant" here, in which case I would agree.

The authors must perform LD isolation

the reviewer is incorrect to say that this must be done. It is a huge amount of extra work. Furthermore, it is not at all necessary when the protein in question localises clearly to the structures, and its may not even work as the protein may need a reasonably high general concentration to avoid gradual dissociation (with any re-association) during organelle purification.

what features of the protein enable its LD binding?

Here an Amphipathic Helix has been identified, but the lack of dissection of that region by the authors explains this question by the reviewer, which is also shared by Reviewer 1. I agree with the implication that more should be done to dissect that.

Major Point 2: Quantitative image analysis:

Mander's Colocalization analyses with Costes correction are required

No. The images in Figure 4 speak for themselves.

Please show the LD phenotype of untransfected, and CbEPF1-GFP transfected cells also

This is a good idea.

provide a means to quantify the clustering of LDs

Unnecessary. Not all findings need to be quantified.

Major Point 3:

Data depends largely on overexpression of the protein in uninfected cells.

I agree

What is the localization of the protein in infected cells?

I wondered about the same point, but I concluded that this is not likely to be possible in the scope of this submission. If it were possible then I guess the authors would have attempted it. Looking on Google Scholar I could find no example of an endogenous Coxiella proteins being tagged in the bacterial genome. So the only way to find the portion is via an antibody. Assuming the authors do not have one, I do not think we should ask for one at this stage in the publication process.

What happens to ER-LD contacts upon infection with C. burnetii?

This is a very valid question, and answering it would not only strengthen the manuscript but should be achievable in 1-3 months.

Reviewer #2 (Significance (Required)):

This work takes a reasonably big step towards uncovering how parasites have mimicked the molecular machinery of contact sites, here in the context of ER-LD interactions and tantalizingly suggestive of lipid transfer at that contact site (although hard to get strong evidence for that at this stage). This provides yet more evidence for the conservation and overall importance to cells of lipid transfer at contact sites, as well as reminding us of the ability of parasites to attack every aspect of cell function.

Reviewer #3 (Evidence, reproducibility and clarity (Required)):

Angara et al describe a secreted *Coxiella burnetii* effector with an FFAT motif, which they name CbEPF1, that localizes to host cell LDs and is able to interact with VAP domain containing host proteins. Furthermore, this interaction is able to impact LD size. The study has several interesting findings that suggest that this protein can impact ER structuring around lipid droplets. The use of bacterial 2-hybrid approach to demonstrate binding between CbEPF1 and VAP domain containing proteins is convincing, further supported by co-ip experiments that go on to establish specificity of each of the FFAT motifs for distinct VAP domain containing proteins. However, the relationship between LD localization of the protein and ER-restructuring to the outcome of infection is not clear from the data presented here. In addition, there are many analyses that need to be performed for the data to convincingly support the claims made in the manuscript.

My major concerns are:

1. The authors claim that CbEPF1 localizes to lipid droplets in *Coxiella burnetii* infected epithelial cells. Towards this, the authors express CbEPF1-GFP in *C. burnetii* infected cells expressing BFP-KDEL. The data in Figure 1B and 1C indicate that the protein localizes to lipid droplets in oleic acid treated cells. It is interesting to note that without addition of oleic acid, the protein localizes to the ER. What is surprising is that the BFP-KDEL signal is also localizing to the LD surface (Figure 1B and 2A).

Both Fig. 1B and 1C were conducted in the absence of oleic acid treatment. In the absence of oleic acid treatment, we observed CbEPF1 localization at the ER, LD biogenesis sites and the LD surface (Fig. 1C). While oleic acid was used to increase the number of LDs in Fig. 2A and 2B, even without oleic acid treatment, CbEPF1-GFP localizes to LD surface (Fig. 1B and 1C).

BFP-KDEL was used as a marker for ER in all our experiments. We only see BFP-KDEL localization around lipid droplets when EPF1 is present. Our data does not suggest that BFP-KDEL itself is on the LD surface.

While in this section it seems that the protein migrates from ER to LDs, in the later sections, similar data is used to make the claim that the protein induces ER apposition to the LD and localizes to regions of LD-ER contact. Therefore, it raises several questions about the localization of the protein:

is it an ER-localized protein that migrates to LDs. In that case, what features of the protein enable its LD binding?

As discussed in detail in the response to reviewers 1 and 2, we discovered an amphipathic helix in the C-terminal region and hydrophobic region in the middle of CbEPF1 are crucial for its LD localization (Fig. 6).

In addition, the authors must perform LD isolation to validate that the protein indeed localizes to lipid droplets.

We believe that we present strong and compelling evidence that CbEPF1 is present on LDs. While outside the scope of the current study, we plan to isolate LDs from CbEPF1-expressing cells for lipidomic analysis and will also include proteomic analysis.

Is it an ER-localized protein, that like BFP-KDEL has the ability to localize to ER-LD contact sites, but remains on the ER membrane? Again, biochemical evidence supporting membrane specification is important to understand the localization of the protein. The authors have focussed only on the FFAT motifs of CbEPF1 and not described the overall domain analysis of the protein. It is therefore difficult to understand at this point how the protein localizes to the ER and the ER-LD junction or LD.

Because there are no readily identifiable domains outside of the FFAT motifs and amphipathic helix, we performed truncated mutant analysis to identify signals for CbEPF1 localization. We found that the hydrophobic region (101-212 amino acids) in the middle of CbEPF1 and an amphipathic helix in the C-terminal region (262-279) are crucial for CbEPF1 localization to LDs (Fig. 6).

2. Quantitative image analysis:

(i) Authors must perform colocalization analyses to substantiate the claims for ER/LD localization. The authors refer to "extended ER-LD contacts" in figure 2B and the text. These data need to be supported with colocalization analysis between BFP-KDEL and the GFP channel.

We have added colocalization analysis as a figure EV1.

(ii) Related to Figure 4A: Mander's Colocalization analyses with Costes correction are required to convincingly demonstrate that the dual FFAT motif is required for ER-LD contacts.

While we appreciate the reviewer's comment, we feel that the Figure 4A, combined with Figure 4B, clearly demonstrate that the ER is no longer wrapped around LDs in the F3 mutant.

(iii) Related to Figure 4B: Please show the LD phenotype of untransfected, and CbEPF1-GFP transfected cells also. Can the authors provide a means to quantify the clustering of LDs.

Control images for GFP and CbEPF1-GFP have been added to Fig 4C of the revised manuscript, while the LD phenotype of CbEPF1-F1mt-GFP and CbEPF1-F2mt-GFP shown in Figure EV3.

In regard to quantitation of LD clustering, we have not devised a method that would provide new information and feel that the images clearly demonstrate the phenotype.

(iv) Figure 5A and B. It is not clear from the figure legend whether the data are pooled from multiple experiments or a single experiment. Experimental replicates must be incorporated in the final analysis.

Thank you for pointing out this oversight; additional experimental details are now included in the revised manuscript, lines 707-719.

3. The data presented in this manuscript depends largely on overexpression of the protein in uninfected cells. Given that C. burnetii induces LD formation, there are three main areas that need clarity:

(i) What is the localization of the protein in infected cells without the addition of oleic acid under conditions where infection itself induces LD biogenesis.

Figure 1B was from an experiment conducted on infected cells, without addition of oleic acid. In this figure, CbEPF1 localizes to both the ER and LDs.

(ii) What happens to ER-LD contacts upon infection with C. burnetii?

We analyzed ER-LD contacts in *Coxiella*-infected macrophages using transmission electron microscopy (TEM). We found *Coxiella* infection significantly increases ER-LD contacts in the host cells compared to the uninfected cells; this data is presented as Fig. 5A and B of the revised manuscript.

(iii) Does the presence of CbEPF1 play any role in infection induced LD biogenesis? This may be an optional experiment to undertake at this point as this would involve significant amount of time investment in generating bacterial strains.

This is an important question we will be addressing in future experiments, as it requires generating mutant bacteria and complements.

Minor comments:

1. Line 36: "maintain" instead of "maintains"

Thank you for catching this typo, which has been corrected.

2. The introduction cites mainly reviews with overlapping concepts but does not cite primary literature in the area of organelle-organelles contacts and inter-organelle communication (lines 39-40 and line 49). It would be good to cite key primary research articles in these areas.

Thank you for the suggestion. We have included the primary research articles that first showcased contacts between each cell organelle (lines 39-41 and lines 50-52), in the revised manuscript.

3. There are some crucial references related to bacterial secreted effectors that target host lipid droplets, that are missing from the introduction. For example, Chlamydia trachomatis is known to secrete effectors that localize to host lipid droplets (PMID: 18591669). Legionella pneumophila secretes a small GTPase LegA15 to lipid droplets impacting vesicle secretion of the host cell (PMID:36525490).

We have discussed the lipid droplet-associated bacterial effectors in the discussion section, in the revised manuscript (lines 320-322).

4. For all figures, please show all individual channels in monochrome and a merge of BFP-KDEL+LD marker and merge of CbEPP1-GFP+LD marker and/or BFP-KDEL+CbEPP1 (wherever appropriate).

We thank the reviewer for this suggestion. While we also prefer to show individual channels as monochrome, in this manuscript we made a conscious choice for pseudocoloring and overlays to make the figures easily and quickly understood by the reader. We attempted to include the suggested overlays but felt that the figures became too large and confusing.

Reviewer #3 (Significance (Required)):

The study by Angara et al reports a dual FFAT motif containing protein of Coxiella burnetii that impacts ER-LD association. The strengths of the study lie in characterization of the FFAT motifs in VAP domain binding, and the role of these FFAT motifs in mediating ER-LD contacts. However, the claims made towards LD localization versus localization to ER-LD contact sites by this protein are not well supported by the data. In addition, the relevance of these findings in infected cells are not addressed in this study as the data presented here pertain to an over-expression system.

Bacterial proteins that exit the bacterial containing vacuole and impact organelles outside the endocytic compartments are fascinating as they have the potential to impact global processes such as transcription, metabolism, and protein secretion. Lipid droplet (LD) homeostasis is dysregulated in many bacterial infections and also plays a crucial role in host defense against infection. Therefore, the knowledge of bacterial effectors in this dysregulation will also potentially provide means of countering bacterial strategies to affect host LD homeostasis. Therefore, the findings presented by Angara et al will provide conceptual and mechanistic advances to the specialized audience in infection and immunity. However, I can foresee that the findings have the potential for interesting tools to be developed for organelle-organelle contacts to be studied, provided there is more clarity on how the protein localizes to the ER/ER-LD contacts/LDs.

Dear Dr. Gilk

Thank you for the submission of your revised manuscript to EMBO reports. We have now received the full set of referee reports that is copied below.

As you will see, all referees are very positive about the study and request only minor changes that should be addressed in the text and in a point-by-point response.

From the editorial side, there are also a few things that we need before we can proceed with the official acceptance of your study.

- Please reduce the number of keywords to 5.
 - Please remove the DOI numbers from the reference list. DOIs are only used for preprint/bioRxiv citations.
 - Please add the header "DISCLOSURE AND COMPETING INTERESTS STATEMENT" to your COI declaration. For more information see <https://www.embopress.org/page/journal/14693178/authorguide#conflictsofinterest>
 - Regarding the Author Contributions, we now use CRedit to specify the contributions of each author in the journal submission system. Therefore, please remove the Author Contributions from the manuscript file and make sure that the author contributions in our online manuscript tracking system are correct and up-to-date. The information you specified in the system will be automatically retrieved and typeset into the article. You can enter additional information in the free text box provided, if you wish.
 - We perform a routine image integrity check on all manuscripts before acceptance. In this case we noticed that the image shown in Figure 2B, lower row, merged, seems to have been reused in Figure EV1. Please clarify and clearly indicate the re-use in the figure legend, even if the image is only used for illustration purposes to explain a method.
 - Data availability statement: please add the specific URLs for S-BIAD1302 and S-BIAD1300 datasets and ensure that the data are accessible.
 - Our production/data editors have asked you to clarify several points in the figure legends (see below). Please incorporate these changes in the manuscript and return the revised file with tracked changes with your final manuscript submission.
- A) Statistical test information. Only p-values that are actually shown in the figure panel(s) should (and must) be defined in the legends, all others should be removed from (or added to) the legend. Moreover, we ask for the specification of exact p-values:
- Please note that the exact p values are not provided in the legends of figures 5b; 7a-b.
 - Please note that in figures 7a-b; there is a mismatch between the annotated p values in the figure legend and the annotated p values in the figure file that should be corrected.

B) Replicates and error bars: Please note that information related to n is missing in the legend of figure 5b.

C) Data presentation: Please note that the scale bar needs to be defined for figures 7c; EV 2.

- The section order should be corrected: title page with complete author information, abstract, keywords, introduction, results, discussion, methods, data availability section, acknowledgements, disclosure and competing interests statement, references, main figure legends, tables, Expanded view figure legends.
- The paragraph "MATERIALS & CORRESPONDENCE" section should be removed
- As a standard procedure, we edit the title and abstract of manuscripts to make them more accessible to a general readership. Please find my suggestions below my signature.
- On a different note, I would like to alert you that EMBO Press offers a new format for a video-synopsis of work published with us, which essentially is a short, author-generated film explaining the core findings in hand drawings, and, as we believe, can be very useful to increase visibility of the work. This has proven to offer a nice opportunity for exposure i.p. for the first author(s) of the study. Please see the following link for representative examples and their integration into the article web page:
https://www.embopress.org/video_synopses
<https://www.embopress.org/doi/full/10.15252/emboj.2019103932>

With kind regards,

Referee #1:

The new experiments on the AH add value to the manuscript. However the dissection of residues 1-212 into 2 halves is flawed. One further point: I persist in believing that some mention should be made of the AlphaFold2 predictions for CBU1370 and related proteins - this is the one place to mention them.

Details:

Line 99: [CBU1370] has no other identifiable conserved domains or motifs.

I think that this is an overstatement, as AF2 produces a model of a globular domain made of helices at the N-terminus. On top of that, as I wrote before, a bacterial homolog (that is FFAT +ve) contains the same helical N-terminal domain and there the prediction is largely believable (pLDDT{greater than or equal to}70%), making the prediction for CBU1370 also believable, and hence reportable. So this sentence in the Introduction should be toned down and a sentence added to the Discussion.

"middle hydrophobic domain": the comment above is important not only in its own right but also in relation to the "middle hydrophobic domain". The structural prediction for A0A1J8NR10_9COXI indicates that the boundaries used here (101-212) are likely to have severed a functional domain in half. 101-212 contains 3 predicted helices, while 1-100 contains 4 predicted helices. In the conserved structure (as found by AF2) these two halves interact intimately. It is entirely believable that severing the two parts of this protein artefactually exposes an internal (hydrophobic) surface by which 101-212 normally contacts 1-101. This almost entirely invalidates the experiments in Fig 6F/G. However, this section is not essential to make the most important point about CBU1370, namely that there is a second means by which is targets LDs apart from the identified AH. Also on this general point, I suggest that the representation of CBU1370 in Fig 8 should be changed to better reflect the AF2 prediction: the part binding VAP is should be "thin" (linear), and the part binding the LD should be round (globular).

The authors could cement the AH's role in LD targeting by showing the % of LDs targeted in images similar to 6B compared to 6C.

Minor point: I notice that there are few CDs in Fig 6E. Could a better image be found?

Referee #2:

The revised manuscript is significantly improved and the authors have addressed my major concerns.

Only one minor point: While the authors demonstrate interaction with VAP domain proteins, the possibility of the wild type CbEPF1 (line 293) or Fmt3 (line 207) engaging in homodimerization is highly speculative and should be removed at this point in the absence of any data in this manuscript pointing towards that.

Referee #3:

The authors have adequately addressed my concerns and answered my questions. I have not additional comments. This is an elegant and exciting study that will be of interest to both the host-pathogen interaction and the cell biology communities.

ABSTRACT:

Effective intracellular communication between cellular organelles occurs at dedicated membrane contact sites (MCSs). Tether proteins are responsible for the establishment of MCSs, enabling direct communication between organelles to ensure organelle

function and host cell homeostasis. While recent research has identified tether proteins in several bacterial pathogens, their functions have predominantly been associated with mediating inter-organelle communication between the bacteria containing vacuole (BCV) and the host endoplasmic reticulum (ER). Here, we identify a novel bacterial effector protein, CbEPF1, which acts as a molecular tether beyond the confines of the BCV and facilitates interactions between host cell organelles. *Coxiella burnetii*, an obligate intracellular bacterial pathogen, encodes the FFAT motif-containing protein CbEPF1 which localizes to host lipid droplets (LDs). CbEPF1 establishes inter-organelle contact sites between host LDs and the ER through its interactions with VAP family proteins. Intriguingly, CbEPF1 modulates growth of host LDs in a FFAT motif-dependent manner. These findings highlight the potential for bacterial effector proteins to impact host cellular homeostasis by manipulating inter-organelle communication beyond conventional BCVs.

Editor Comments

Please reduce the number of keywords to 5.

Keyword number reduced to 5.

Please remove the DOI numbers from the reference list. DOIs are only used for preprint/bioRxiv citations.

DOI numbers were removed from the reference list.

Please add the header "DISCLOSURE AND COMPETING INTERESTS STATEMENT" to you COI declaration. For more information see <https://www.embopress.org/page/journal/14693178/authorguide#conflictsofinterest> [embopress.org]

Statement header was included in the revised manuscript.

Regarding the Author Contributions, we now use CRediT to specify the contributions of each author in the journal submission system. Therefore, please remove the Author Contributions from the manuscript file and make sure that the author contributions in our online manuscript tracking system are correct and up-to-date. The information you specified in the system will be automatically retrieved and typeset into the article. You can enter additional information in the free text box provided, if you wish.

Author contributions were removed from the manuscript file and checked for accuracy in the journal submission system.

We perform a routine image integrity check on all manuscripts before acceptance. In this case we noticed that the image shown in Figure 2B, lower row, merged, seems to have been reused in Figure EV1. Please clarify and clearly indicate the re-use in the figure legend, even if the image is only used for illustration purposes to explain a method.

A statement clarifying the re-use was included in the figure EV1 legend, along with clear mention of re-use of the figure 2B merged image from bottom panel.

Data availability statement: please add the specific URLs for S-BIAD1302 and S-BIAD1300 datasets and ensure that the data are accessible.

URLs for each dataset have been added to the data availability statement.

Our production/data editors have asked you to clarify several points in the figure legends (see below). Please incorporate these changes in the manuscript and return the revised file with tracked changes with your final manuscript submission.

A) Statistical test information. Only p-values that are actually shown in the figure panel(s) should (and must) be defined in the legends, all others should be removed from (or added to) the legend.

Moreover, we ask for the specification of exact p-values:

- Please note that the exact p values are not provided in the legends of figures 5b; 7a-b.
- Please note that in figures 7a-b; there is a mismatch between the annotated p values in the figure legend and the annotated p values in the figure file that should be corrected.

Exact *P* values are now provided in the figures 5b and 7a, and in the respective figure legends. In Figure 7b, the p value is beyond the maximal limit to view in Prism (15 digits, $P < 1 \times 10^{-15}$).

B) Replicates and error bars: Please note that information related to n is missing in the legend of figure 5b.

Information related to 'n' was included in the revised manuscript. We noticed 'n' was wrongly defined as 'number of experiment repeats' instead of 'sample size' in the figure legends 7A and 7B. We corrected it in the revised manuscript.

C) Data presentation: Please note that the scale bar needs to be defined for figures 7c; EV 2.

Scale bars defined in the respective figure legends

The section order should be corrected: title page with complete author information, abstract, keywords, introduction, results, discussion, methods, data availability section, acknowledgements, disclosure and competing interests statement, references, main figure legends, tables, Expanded view figure legends.

Completed.

MATERIALS & CORRESPONDENCE" section should be removed

Removed.

As a standard procedure, we edit the title and abstract of manuscripts to make them more accessible to a general readership. Please find my suggestions below my signature.

We incorporated your suggestions in the revised abstract.

- On a different note, I would like to alert you that EMBO Press offers a new format for a video-synopsis of work published with us, which essentially is a short, author-generated film explaining the core findings in hand drawings, and, as we believe, can be very useful to increase visibility of the work. This has proven to offer a nice opportunity for exposure i.p. for the first author(s) of the study. Please see the following link for representative examples and their integration into the article web page:

https://www.embopress.org/video_synopses [embopress.org]

<https://www.embopress.org/doi/full/10.15252/emboj.2019103932> [embopress.org]

We would like to provide video synopsis.

Reviewer Comments

Referee #1:

The new experiments on the AH add value to the manuscript. However the dissection of residues 1-212 into 2 halves is flawed. One further point: I persist in believing that some mention should be made of the AlphaFold2 predictions for CBU1370 and related proteins - this is the one place to mention them.

Details:

Line 99: [CBU1370] has no other identifiable conserved domains or motifs.

I think that this is an overstatement, as AF2 produces a model of a globular domain made of helices at the N-terminus. On top of that, as I wrote before, a bacterial homolog (that is FFAT +ve) contains the same helical N-terminal domain and there the prediction is largely believable (pLDDT{greater than or equal to}70%), making the prediction for CBU1370 also believable, and hence reportable. So this sentence in the Introduction should be toned down and a sentence added to the Discussion.

"middle hydrophobic domain": the comment above is important not only in its own right but also in relation to the "middle hydrophobic domain". The structural prediction for A0A1J8NR10_9COXI indicates that the boundaries used here (101-212) are likely to have severed a functional domain in half. 101-212 contains 3 predicted helices, while 1-100 contains 4 predicted helices. In the conserved structure (as found by AF2) these two halves interact intimately. It is entirely believable that severing the two parts of this protein artefactually exposes an internal (hydrophobic) surface by which 101-212 normally contacts 1-101. This almost entirely invalidates the experiments in Fig 6F/G. However, this section is not essential to make the most important point about CBU1370, namely that there is a second means by which is targets LDs apart from the identified AH.

We removed the statement 'has no other identifiable conserved domains or motifs' in the results section (line 99) and included the possibility of globular domain in the N-terminal region of CbEPF1 based on AlphaFold2 predicted structure available in Alpha Fold Protein Structure Database (line 332 – 337).

Also on this general point, I suggest that the representation of CBU1370 in Fig 8 should be changed to better reflect the AF2 prediction: the part binding VAP is should be "thin" (linear), and the part binding the LD should be round (globular).

We removed our model figure (Figure 8) from the main figures to use as a synopsis figure. The revised synopsis figure incorporates the reviewer's suggestions, showing CBU1370/CbEPF1 as a protein with globular domain, amphipathic helix and hydrophobic domain.

The authors could cement the AH's role in LD targeting by showing the % of LDs targeted in images similar to 6B compared to 6C.

We provided the percentage of LDs targeted by CbEPF1-WT (100%) and CbEPF1-AHmt (40%) (line 224-228).

Minor point: I notice that there are few CDs in Fig 6E. Could a better image be found?

Thank you for this suggestion; we replaced Fig. 6E with a better image showing higher LDs.

Referee #2:

The revised manuscript is significantly improved and the authors have addressed my major concerns.

Only one minor point: While the authors demonstrate interaction with VAP domain proteins, the possibility of the wild type CbEPF1 (line 293) or Fmt3 (line 207) engaging in homodimerization is highly speculative and should be removed at this point in the absence of any data in this manuscript pointing towards that.

We removed the mention of possible homo or hetero dimerization from discussion section.

Referee #3:

The authors have adequately addressed my concerns and answered my questions. I have not additional comments. This is an elegant and exciting study that will be of interest to both the host-pathogen interaction and the cell biology communities.

Dr. Stacey D. Gilk
University of Nebraska Medical Center
Department of Pathology and Microbiology
985900 Nebraska Medical Center
DRC2 5031
Omaha, NE 68198
United States

Dear Dr. Gilk,

I am very pleased to accept your manuscript for publication in the next available issue of EMBO reports. Thank you for your contribution to our journal.

Yours sincerely,
